# Enhancement of nanoparticle formation and growth during the COVID-19 lockdown period in urban Beijing

Xiaojing Shen[1,*], Junying Sun[1], Fangqun Yu[2], Ying Wang[1], Junting Zhong[1,3], Yangmei Zhang[1], Xinyao Hu[1], Can Xia[1,4], Sinan Zhang[1], Xiaoye Zhang[1]

5  1 State Key Laboratory of Severe Weather & Key Laboratory of Atmospheric Chemistry of CMA, Chinese Academy of Meteorological Sciences, Beijing 100081, China

2 Atmospheric Sciences Research Center, State University of New York at Albany, 251 Fuller Road, Albany, New York 12203, USA

3 University of Chinese Academy of Sciences, Beijing 100049, China

10  4 Nanjing University of Information Science & Technology, Nanjing 210000, China

*Correspondence to*: Xiaojing Shen (shenxj@cma.gov.cn)

**Abstract**. Influenced by the spread of the global 2019 novel coronavirus (COVID-19) pandemic, primary emissions of particles and precursors associated with anthropogenic activities decreased significantly in China during the Chinese New Year of 2020 and the lockdown period (January 24–February 16, 2020). Two-month measurements of the number size distribution of neutral particles and charged ions showed that during the lockdown (LCD) period, the number concentration of particles smaller than 100 nm decreased by approximately 40% compared to the Pre_LCD period in January. However, the accumulation mode particles increased by approximately 20% as several polluted episodes contributed to secondary aerosol formation. In this study, new particle formation (NPF) events were found to be enhanced in the nucleation and growth processes during the LCD period, as indicated by higher formation rate of 2 nm particles ($J_2$) and the subsequent growth rate ($GR$). The relavent precursors, e.g. $SO_2$ and $NO_2$ showed a clear reduction, $O_3$ increased by 80% during LCD period, as compared with Pre_LCD. The volatile organic vapors showed different trends due to their sources. The proxy sulfuric acid during the LCD period increased by approximately 26%, as compared with Pre_LCD. The major oxidants ($O_3$, OH and $NO_3$) of VOCs were also found to be elevated during LCD. That indicated higher $J_2$ and $GR$ (especially below 5 nm) during the LCD period were favored by the increased concentration level of condensing vapors and decreased condensation sink. Several heavy haze episodes have been reported by other studies during the LCD period; however, the increase in nanoparticle number concentration should also be considered. Some typical NPF events produced a high number concentration of nanoparticles that intensified in the following days to create severe aerosol pollution. Our study confirms a significant enhancement of the nucleation and growth process of nanoparticles during the COVID-19 LCD in Beijing and highlights the necessity of controlling nanoparticles in current and future air quality management.

## 1.   Introduction

As a response to the outbreak of the 2019 novel coronavirus (COVID-19), the Chinese government implemented restrictions

on population movement in February 2020; the period when the restrictions were enforced was also called the lockdown period (LCD). During the LCD period, the $NO_x$ emission was reduced by approximately 50% in China, as retrieved by the satellite (Zhang et al., 2021) and ground-based measurements (Huang et al., 2020). The number concentration of Aitken mode particles (~25-100 nm), which is related with the traffic emission (Deventer et al., 2018) is also expected to decrease. The significant decrease in aerosol and precursor emissions during LCD is associated with reduced human and economic activities. However, several heavy haze pollution periods occurred in the Yangtze River Delta (YRD) and the Beijing–Tianjing–Hebei Province (BTH) region. Secondary particles contribute significantly to air pollution, and $NO_x$ reduction favors increased ozone and atmospheric oxidizing capacity (Huang et al., 2020). The aerosol heterogeneous reaction process was also enhanced by the anomalously high humidity in northern China (Le et al., 2020). Furthermore, particle accumulation could also be favored by stagnant airflow and vertical meteorological conditions during LCD (Zhong et al., 2018).

New particle formation (NPF) has been an active global research topic for the last two decades because of its potential climatic implications (Kulmala et al., 2004). Nucleated particles can reach number concentrations of $10^4$–$10^6$ $cm^{-3}$. Subsequent growth contributes significantly to cloud condensation nuclei (CCN) (Kerminen et al., 2012) and can cause air pollution (Guo et al., 2014). Primary emissions of particulate matter (PM), CO, $SO_2$, and $NO_2$ decreased significantly after the strict clear air action plans were implemented in the last decade by the Beijing government (Zhang et al., 2019). Changes in $SO_2$ and background aerosols, the key factors influencing NPF events, are also linked to the formation ($J$) and growth rates ($GR$) of secondary particles (Kyrö et al., 2014). Nanoparticles (diameter $\leq$ 100 nm) make minor particle mass contributions but pose a serious risk to human health because of high number concentrations and deep respiratory and cardiovascular system penetration (Kawanaka et al., 2009). However, the size-resolved, chemical, and toxicological properties of nanoparticles are unclear (Jin et al., 2017). Under unfavorable meteorological conditions, the growth of the nanoparticles for several consecutive days would even probably lead to particle mass enhancement as found in Beijing, China (Guo et al., 2014).

In a previous study, the influence of NPF event occurrence by emission reduction in Beijing was analyzed for China's Victory Day parade in August 2015 and for the summer Olympics in 2008; during this period, higher NPF occurrence frequency but lower $J$ and $GR$ was reported as a result of low precursor concentrations (Shen et al., 2016). In the present study, we focus on changes in particle number size distribution and NPF events during LCD in Beijing and the influencing factors. The link between NPF events and regional aerosol pollution is also explored. Our study will facilitate the optimization of regulatory measures to control particle and gas pollution in China, especially with regard to the variation of NPF-associated condensing vapors caused by reduced precursor emissions and elevated atmospheric oxidizing level.

## 2. Methods

### 2.1 Measurements

The particle and ion number size distribution measurements were conducted in January and February 2020 on the roof of the Chinese Academy of Meteorological Sciences building (CAMS) on the Chinese Meteorological Administration campus. The site is approximately 53 m above ground level and located in the western Beijing urban area between the second and third ring roads. A major road with heavy traffic to the west of the site indicated that the sample air could be influenced by traffic emissions. More information about the site can be found in X. Wang et al. (2018) .

### 2.2 Instrumentation

The number of particles of sizes 10–850 nm was measured using a scanning mobility particle sizer (SMPS, TROPOS, Germany). The system is a combination of a differential mobility analyzer (DMA) and a condensation particle counter (CPC, Model 3772, TSI Inc., USA). The mobility distribution of naturally charged and neutral nanoparticles is measured by a neutral cluster and air ion spectrometer (NAIS) with a 10-min time resolution (Mirme et al., 2007, Mirme and Mirme, 2013). The measured mobility was in the range 3.3–0.0013 $cm^2\,V^{-1}\,s^{-1}$, corresponding to mobility diameters of 0.8–42 nm. Positive and negative ions were simultaneously classified by two cylindrical differential mobility analyzers (DMAs) and detected with 21 electrometers on the outer cylinder. A high sample flow rate of 60 lpm was used to minimize diffusion losses. In ion mode, the detected signal is inverted to a mobility distribution consisting of 28 bins, taking into account the measured background and experimentally determined diffusion losses. In neutral particle mode, the sample aerosol is charged using corona chargers, and the charged fraction is calculated (Fuchs, 1963). However, the lowest detection limit for the NAIS in the neutral particle mode (approximately 2 nm) was affected by the corona-generated ions (Asmi et al., 2009, Manninen et al., 2011). The lowest detection limit of the NAIS in ion mode was determined by the charging probability, nanoparticle concentration, and charger ion mobility (Kulmala et al., 2013).

Volitale organic vapors (VOCs) were measured using a Proton Transfer Reaction-Time of Flight Mass Spectrometery (PTR-ToF-MS 8000, IONICON) with a hydronium ion ($H_3O^+$) source at CAMS site. About 30 kinds of compounds could be detected by the PTR-ToF-MS, by using the linear regression multipoint calibrations (Yuan et al., 2017). In this study, mixing ratio of isoprene and major C6–C9 VOCs were derived with 1 h time resolution, which are good indicators of anthropogenic VOC plumes (Dai et al., 2017).

The mass concentrations of $PM_{2.5}$, precursor gases of $O_3$, $SO_2$, $NO_2$, and CO at the GuanYuan air quality monitoring site were derived from the data center of the Ministry of Ecology and Environment of the People's Republic of China (http://datacenter.mep.gov.cn), which is 3 km from the CAMS site. The global radiation at the observatory (54511) in the

southern Beijing urban area was used to estimate the sulfuric acid in this work. The solar radiation datasets were provided by

the National Meteorological Information Center of the China Meteorological Administration. The meteorological factors used

in this study—wind speed (WS), wind direction (WD), and relative humidity (RH)—were derived from the Haidian National

Basic Meteorological Station (54399). The data can represent meteorological conditions at the CAMS site, which is located

~5 km northwest of the urban area site.

## 2.3 NPF parameter calculation

The total particle and ion formation rate at 2 nm ($J_{2,tot}$ and $J_{2,ion}$) can be calculated from the particle and ion number

concentrations in the size range 2–3 nm (Hirsikko et al., 2011, Manninen et al., 2010). $J_{2,tot}$ and $J_{2,ion}$ included time changes in

the concentration of 2–3 nm particles or ions (first term on the right side of Eq. 1 and 2), coagulation loss of 2–3 nm particles

or ions with the pre-existing particles derived by SMPS (second term), and growth of 2–3 nm particles or ions into larger sizes

by the growth rate, $GR$ (third term). In Eq. (2), the fourth and fifth terms represent loss due to ion–ion recombination and

formation from ion-neutral attachment. The equations are given by the following formulas:

$$J_{2,tot} = \frac{dN_{2-3,tot}}{dt} + CoagS_2 \times N_{2-3,tot} + \frac{GR_3}{1nm} \times N_{2-3,tot} \tag{1}$$

$$J_{2,ion}^{\pm} = \frac{dN_{2-3,ion}^{\pm}}{dt} + CoagS_2 \times N_{2-3,ion}^{\pm} + \frac{GR_3}{1nm} N_{2-3,ion}^{\pm} + \alpha \times N_{2-3,ion}^{\pm} N_{<3,ion}^{\mp}$$

$$- \beta \times N_{2-3,par} N_{<2,ion}^{\pm} \tag{2}$$

$N_{2-3,tot}$ and $N^{\pm}_{2-3,ion}$ are the number concentration of particles and ions of positive and negative charges, respectively. $CoagS_2$ is

the 2 nm coagulation coefficient. $\alpha$ and $\beta$ (the ion–ion recombination and ion-neutral attachment coefficients, respectively)

are assumed to be $1.6\times10^{-6}$ and $10^{-8}$ cm$^3$s$^{-1}$, respectively, (Hoppel, 1985).

Growth rate ($GR$) is defined as the rate of change of diameter with time, $GR = (D_{p,2} - D_{p,1})/dt$, given in nm h$^{-1}$, where $D_{p,1}$

and $D_{p,2}$ are the geometric mean diameters (GMD) when the nucleated particles start and stop growing, respectively. GMDs

are derived by the log-normal modal fitting of the particle/ion number size distributions (Hussein et al., 2009).

Sulfuric acid ($H_2SO_4$) is a key component in the nucleation process (Kulmala et al., 2013). The concentration of $H_2SO_4$ was

not measured directly in this study and different proxy methods were refered to derive the proxy sulfuric acid. A method (Eq.

3) depends on the global radiation (Glob_R), $SO_2$ and condensation sink ($CS$), and is developed according to the previous

study conducted in a forest site, Hyytiälä, Finland (Petäjä et al., 2009).

$$[H_2SO_4] = \frac{k \times Glob\_R \times [SO_2]}{CS} \tag{3}$$

where $k$ is empirically derived factor and well correlated with Glob_R ($k=1.4\times10^{-7}\times$Glob_R$^{-0.7}$, unit: m$^2$ W$^{-1}$ s$^{-1}$). The proxy

equation is site-specific due to the different atmospheric comditions. In the polluted atmosphere, such as in Beijing, several

proxy methods were also constructed based on a number of available atmospheric parameters (Lu et al., 2019). In this study,

the simplext proxy (Eq. 4) and best performed proxy (Eq. 5) in Lu et al. (2019) are adpoted to derive the proxy sulfuric acid.

$$[H_2SO_4] = 280.05 \times UVB^{0.14} \times [SO_2]^{0.40} \tag{4}$$

$$[H_2SO_4] = 0.0013 \times UVB^{0.13} \times [SO_2]^{0.40} \times CS^{-0.17} \times ([O_3]^{0.44} + [NO_x]^{0.41}) \tag{5}$$

[H$_2$SO$_4$] is the gaseous sulfuric acid with the unit of molecule cm$^{-3}$. [SO$_2$], [O$_3$] and [NO$_x$] is the concentration of sulfur dioxide, ozone, and nitrogen oxides, with the unit of molecule cm$^{-3}$. UVB is the intensity of ultraviolet radiation $b$ in W m$^{-2}$. $CS$ is the condensation sink, which describes how fast the vapor molecules condense on the existing particles (Dal Maso et al., 2002), with the unit of s$^{-1}$. The proxy method has been validated by comparing the measured sulfuric acid with a high correlation coefficient of 0.86 (Lu et al., 2019), based on the field campaign conduted approximately 2 km away from CAMS site. Although the direct measurement of UVB was not available, it had been reported by Hu et al. (2013) that the monthly average of the ratio of UVB to global radiation (Glob_R) ranged from 0.007 to 0.017% in Beijing. And in this study, the average ratio of January and Feburary (0.008%) was applied to to derive UVB by 0.008%$\times$ Glob_R. The covariance of $CS$ and SO$_2$ was found (correlation coefficient $R$=0.83) that offset the dependence of sulfuric acid on $CS$ by Lu et al. (2019). However, the anthropogenic emission sharply decreased during LCD in this study, and $R$ was 0.45 for SO$_2$ and $CS$. To minimize the uncertainty of H$_2$SO$_4$ proxy, the average value of three calculation methods was applied for the further analysis.

## 2.4   Typical NPF event identification

NPF events are identified and different nucleation types are characterized based on the daily evolution of particle number size distribution (PNSD). The burst of nucleation mode particles with diameter $\leq 25$ nm appeared in the PNSD, and the burst should prevail over a few hours with clear growth process (Dal Maso et al., 2005). Regional NPF events can occur over a geographically large area and extend over several hundreds of kilometers (Shen et al., 2018). Such events indicate regional cases in which freshly nucleated particles can reach the size of CCN (Shen et al., 2011).

## 2.5   Back trajectory analysis

In order to reveal the meteorological condition during the pollution case formation, the 48 h backward trajectories arriving at CAMS stie were calculated at 12:00 Local time during February 4-14 for case study, terminating at the height of 500 m above ground level by applying the Trajstat Software, combined with HYSPLIT 4 model (Hybrid Single-Particle Lagrangian Integrated Trajectory) and using the NCEP GDAS (Global Data Assimilation System) data with 1°×1° resolution (Draxler and Hess, 1998, Wang et al., 2009).

## 3. Results and discussion

### 3.1 The meteorological conditions

The meteorological parameters during LCD period, January and February in 2020, as well as the average conditions of January and February in 2016-2020 were analyzed and the diurnal pattern was given (Fig. 1). It showed much higher RH, lower WS, slightly higher T and lower pressure during LCD, January and February 2020, than that of 5-year climatology average condition (January and February in 2016-2020). The anomaly of monthly mean sea level pressure in January and February between 2020 and 2016-2020 was analyzed based on the ECMWF reanalysis dataset (ERA5, https://cds.climate.copernicus.eu/), as given in the supplementary materials (Fig. S1). It showed negative anomaly in BTH region, indicating the air pressure decreased in January and February in 2020, as compared with the corresponding period of the 5-year climatology. The local air convergence resulted in high RH and low WS, which favored for the air pollutants accumulating (Zhong et al., 2018). The unfarable meteorological trapped moisture and pollutants near the ground, thus could offset substantial emissions reductions during COVID-19 LCD to some extent.

### 3.2 Overview of the NSD of particles and charged ions

Fig. 2 shows the time evolution of the number size distribution (NSD) of particles in the 10–850 nm range, neutral particles (2–42 nm), and charged ions (0.8–42 nm) in January and February 2020. The dataset was classified into the COVID-19 LCD (January 24–February 16, 2020), Pre_LCD period (January 3–23, 2020), and Post_LCD (February 17–29, 2020) to reveal the influence of emission reductions. The NPF event occurred on 10 out of 23 days during Pre_LCD, 10 out of 24 days in LCD, and 5 out of 13 days in Post_LCD, respectively. Poisson statistics was conducted for NPF event occurrence probability for Pre_LCD, LCD and Post_LCD period, respectively, as given in Fig. S2 in supplementary materials. It showed slight variation of NPF event occurrence probability, as compared with Pre_LCD and LCD period. Despite large primary emissions reduction, several cases of heavy aerosol pollution events occurred in the BTH region during LCD. Particle matter below 2.5 μm (PM$_{2.5}$) mass concentration at air monitoring sites in Beijing of the Ministry of Ecology and Environment of China exceeded 75 μg/m$^3$ (the second grade of the Ambient Air Quality Standard of China) on 12 of the 28 days, which were identified as polluted conditons. The elevated PM mass concentration was attributed to the secondary aerosol formation process; this process was aided by the enhanced oxidizing capacity caused by increased ozone levels (Huang et al., 2020).

The particle number concentrations of the Aitken mode (25–100 nm, N$_{25-100nm}$) and the accumulation mode (100–850 nm, N$_{100-850nm}$) derived by SMPS and the nucleation mode ($\leq$ 25 nm) of neutral particles and charged ions by NAIS were given in Fig. 3 and discussed in detail in the following. The Aitken mode showed a significant reduction since the Chinese New Year (January 24) and normal fluctuations below 3000 cm$^{-3}$ during LCD and Post_LCD. Mean N$_{25-100}$ concentrations were 4040

± 1590, 2400 ± 1170, and 2170 ± 994 cm$^{-3}$ in Pre_LCD, LCD, and Post_LCD, respectively. Aitken mode particles were closely related to the anthropogenic emissions and reduced by approximately 40%. During Post_LCD, the Aitken mode concentration remained low because people were encouraged to work at home and services were almost shut down. Accumulation mode particles usually undergo coagulation, condensation, heterogeneous reactions, and long-range transport processes that can reflect regional polluted conditions. $N_{100-850nm}$ concentrations were 1820 ± 1190, 2200 ± 1320, and 1850 ± 840 cm$^{-3}$ during Pre_LCD, LCD, and Post_LCD, respectively; the 20% increase during LCD (compared with Pre_LCD) occurred despite large emissions reductions and was related to specific pollution episodes that occurred from January 24–26 and February 12–14. The particle number concentration derived from SMPS is probably lower than that from NAIS in the overlap size range of 20–40 nm, because the overestimation of natural particle concentration as a multiple charge effect above 20 nm is beyond the instrumental detection limit (Gagné et al., 2011). In this study, the number concentration of 20–40 nm was integrated by the SMPS and NAIS particle mode ($N_{20-40nm, smps}$ and $N_{20-40nm, nais}$) with an enhancement factor ($N_{20-40nm, nais}/N_{20-40nm, smps}$) of 1.65 ± 0.06, and the number concentration of particles larger than 20 nm derived by SMPS was more accurate.

The nucleation mode ($N_{par, \leq 25\ nm}$) derived from NAIS was separated into ≤ 10 nm and 10–25 nm for neutral particles ($N_{par,nais,2-10nm}$, $N_{par,nais,10-25nm}$) and positively charged ions ($N_{ion,nais,1-10nm}$, $N_{ion,nais,10-25nm}$), respectively. $N_{par,nais,2-10nm}$ was the primary contributor to the nucleation mode, which was determined by NPF events, during which average peak $N_{par,nais,\leq 25\ nm}$ concentrations were 2.3 ± 2.3×10$^4$, 1.5 ± 2.6×10$^4$, and 1.9 ± 3.3×10$^4$ cm$^{-3}$, during Pre_LCD, LCD, and Post_LCD, respectively (Fig. 3b). The number concentration of 10–25 nm particles could also be derived from SMPS ($N_{par,smps,10-25nm}$), which was also given in Fig. 3b and approximately 30% lower than the value of $N_{par,nais,10-25nm}$. $N_{par, \leq 25\ nm}$ showed large variation because of significant differences between NPF and non-NPF days. However, several cases during LCD showed a significantly high peak $N_{par,2-10nm}$ value (Fig. 3c), indicating the probability of the stronger nucleation process during LCD. The positive and negative ion number concentrations of 0.8–42 nm were 457 ± 245 and 496 ± 265 cm$^{-3}$, respectively. The mean values of $N_{ion,nais,1-10nm}$ and $N_{ion,nais,10-25nm}$ ranged from 100–200 cm$^{-3}$, indicating a minor contribution to the total particle count.

### 3.3 NPF event variation

Table 1 provides the key parameters describing NPF events, including NPF days and available measurement days, $CS$, $J_2$, and $GR$ for total particles and charged ions. Higher $J_2$ and $GR$ values for particles and ions were also found during LCD and Post_LCD than during Pre_LCD. However, the emissions control period during the China Victory Parade in Beijing in 2015 (August 20–September 3) featured higher frequency, decreasing $J_3$ and $GR$ trends compared with the corresponding month in 2010–2013 (Shen et al., 2016). $J_3$ referred the formation rate at 3 nm calculated from the particle number concentration of 3–4 nm particles by Eq. (1), as the lowest detection limit of SMPS applied in 2015 and 2010-2013 campaign was 3 nm. That indicated the factors influencing the NPF event, including precursors, pre-existing particles and meteorological conditions,

were complex and should be evaluated further. The daily mean value of $NO_2$ decreased by ~35% and $SO_2$ decreased by ~13%, whereas $O_3$ increased by 80% during LCD as compared to Pre_LCD in this work (Fig. 4). The probability density function (PDF) was analyzed for hourly $SO_2$, $NO_2$ and $O_3$ during Pre_LCD, LCD and Post_LCD, respectively and the result was given in the supplementary materials (Fig. S3). It also showed significant decreasing trend of $NO_2$, whereas increasing trend of $O_3$ as compared with Pre_LCD and LCD/Post_LCD. However, the variation of $SO_2$ among different periods was not clear, as the $SO_2$ concentration remained low due to the emission control these years. Previous studies had indicated that $NO_x$ suppressed NPF events by influencing the formation of highly oxygenated organic molecules (HOMs), which participated in nucleation and initial particle growth (Lehtipalo et al., 2018, Yan et al., 2016, 2020), suggesting that the reduction of $NO_2$ during LCD provided favorable conditions for particle growth.

In this work, five kinds of VOCs (isoprene, benzene, toluene, C8 and C9 aromatics) are discussed, which are the indicators of anthropogenic VOC and also could be oxidized to be HOMs to contribute to the growth process (Dai et al., 2017). The result (Fig. 5) showed C8 and C9 aromatics decreased by approximately 20% and 8% during LCD as compared with Pre_LCD, however, isoprene and toluene slightly changed, benzene increased by approximately 21% during LCD period. It also suggested the VOCs we focused didn't show the reduction rate as 45% as Huang et al. (2020) reported in BTH region. The major pathways of HOMs formation are the oxidation by $O_3$, OH and $NO_3$ radicals (Atkinson and Arey, 2003). As mentioned above, $O_3$ increased by 80% during LCD period. We used Glob_R as a simple proxy of OH, and Glob_R increased by ~24% during LCD as compared with Pre_LCD, as given in the supplementary materials (Fig. S4). $NO_3$ oxidation of nocturnal biogenic VOCs is also an important pathway of secondary organic aerosol formation in Beijing (H. Wang et al., 2018). $NO_3$ is predominantly formed by the reaction of $NO_2$ with $O_3$ ($NO_2 + O_3 \rightarrow NO_3 + O_2$ ), and we applied $[NO_2] \times [O_3]$ to estimate the $NO_3$ production. It showed $[NO_2] \times [O_3]$ term increased by ~40% during LCD period. Based on the above discussion, it showed the variations of precursors, solar radiation and $CS$ could finally influence NPF by photochemical reactions with VOCs and sulfuric acid production, promoting the nucleation and growth process.

The $H_2SO_4$ proxy were derived according to equations (3–5) as given in Fig. 6 and the mean value of the three mothds was discussed. For LCD and Pre_LCD during the NPF event occurrence (9:00–16:00 LT), $CS$ decreased by ~25% and Glob_R increased by ~40%, whereas $SO_2$ decreased by ~28%. The variations of these variables finally lead to a $H_2SO_4$ increase of ~26%. The formation of sulfuric acid was aided by the enhanced atmospheric oxidizing capacity because of elevated $O_3$ concentration during LCD, which had also been validated in the previous study in Nanjing, YRD, China (Huang et al., 2020). The $H_2SO_4$ proxy were correlated with $J_{2,tot}$ and $GR$, with the $R$ value of 0.62 for $J_{2,tot}$ and the $H_2SO_4$, and 0.45 for $GR$ and the $H_2SO_4$, respectively. Based on the NAIS data of neutral particle mode, the hourly mean geometric mean diameter of nucleation mode ($D_{p,nuc}$) was fitted to show the growth process and its relationship with proxy $H_2SO_4$ (Fig. 7). It also revealed that in the

initial growth process ($D_{p,nuc} < 5$ nm), $D_{p,nuc}$ increased positively with $H_2SO_4$ proxy. Furthermore, $GR$ in the size range of 3-5

235    nm was slightly higher during LCD and Post_LCD, as compared with Pre_LCD, indicating the enhanced effect of sulfuric

acid on the initial growth of the nucleated particles. However, when the nucleated particles grew into the larger sizes (> 5 nm),

$H_2SO_4$ proxy decreased probably related with the weak solar radiation in late afternoon, which could not explain the continuous

growth and the oxidized VOCs could be the main contributor. The non-linear dependence of $J_{2,tot}$, and $GR$ on the condensing

vapors indicates a complex mechanism in the multi-component nucleation and growth system. Stolzenburg et al. (2020)

revealed that sulfuric acid played an important role in smaller growth processes from 2–10 nm, however, could not explain

condensational growth when the nucleated particles overcame 10 nm. For particles larger than 10 nm, low volatile organic

vapors should contribute to growth (Kontkanen et al., 2018, Yan et al., 2020).

### 3.4    Effect of charged ions

Table 1 provided the parameters describing the nucleation and growth process for neutral particles and positive and negative

ions and showed that the $GR$ of ions was larger than that of neutral particles. Growth enhancement from charge–dipole

interactions between condensable gases and charged ions lead to higher growth rates than with neutral particles (Nadykto and

Yu, 2003, Yu and Turco, 2000). The growth process of $D_{p,nuc}$ of neutral particles and positive ions were given in Fig. 8. It

showed $D_{p,nuc,ion}$ grow faster than $D_{p,nuc,par}$, especially for the sizes below 5 nm, depending on the growth rate in each time

interval (($D_{p,nuc,t1}$- $D_{p,nuc,t2}$)/$\Delta t$, $\Delta t$ = 1 h). The enhanced growth rate factor ($GR_{p,nuc,ion}$/ $GR_{p,nuc,par}$) ranged from 1.1 to 1.7, with

the average of 1.38±0.34 during the entire particle growth process and higher (~2.0) for the initial size of 2–5 nm. The growth

of the nano-sized particles was not linear (especially at the initial size); therefore, the $GR$ calculation was split into different

size ranges (Fig. 9). The $GR$ of charged ions was higher than that of neutral particles for all size ranges, which is consistent

with previous studies (Hirsikko et al., 2005, Suni et al., 2008), and the difference was much larger at initial sizes below 5 nm

as indicated above. In addition to condensational growth, the difference in the loss rates of smaller particles (neutral, positive,

negative) due to the coagulation process and ion–ion recombination also affects particle size and calculated $GRs$ (Yu and Turco,

2008, 2011). The effect of the charge decreases as particle size increases, and more species condense as particles grow.

However, the number concentration of charged ions plays a minor role in the total particle count, and their contribution to the

total growth process and nucleation can be ignored in urban Beijing, where the nucleation mechanism is dominated by neutral

pathway with abundant condensing vapors.

### 3.5    Air pollution episode followed by NPF event

In this study, two severe pollution episodes occurred during LCD from January 24–29 and February 7–14, with daily average

$PM_{2.5}$ mass concentrations in the range 75–210 µg/m$^3$. Both episodes occurred after NPF days on January 23 (No. 10) and

February 4 (No. 16), respectively. Other pollution episodes in January 16–18, February 19–21, and February 28–29 were preceded by NPF events No. 8, 21, and 25, respectively. The most long-lasting pollution episode on February 7–14 is discussed further to reveal the relationship between NPF events and aerosol pollution formation (Fig. 10). The NPF events on February 4–5 produced high concentrations of nucleation mode particles, which grew to 150–200 nm in a few days. Two principal pollution episode formation stages were identified according to variations in the $PM_{2.5}$ mass concentration dividing by CO ($PM_{2.5}/CO$), as indicated in Fig. 10b. The normalized $PM_{2.5}$ by CO represents the secondary aerosol formation effect, which segregates the possible influence of physical effects, such as air mass change and planetary boundary layer (PBL) development (Wiedensohler et al., 2009). The back trajectories arriving at CAMS station at 12:00 local time form February 4 to 14 with the terminal height of 500 m agl were calculated (Fig. 11). The result showed back trajectories originated from northwest from February 4 to 10, corresponding to the dry and clean air masses. However, from February 11 to 13, the southwesterly air masses were dominated and favored the accumulating of the particles, resulting in the high concentration level of $PM_{2.5}$. In the first stage (February 5–10), the secondary aerosol formation was the key process contributing to increasing $PM_{2.5}$ mass. The continuous growth of nucleated particles was intervened by the development of PBL and local wind sometimes. As in the second stage (February 11–13), $PM_{2.5}$ reached a peak value of 250 μg/m$^3$, and $PM_{2.5}/CO$ slightly decreased with small fluctuations. Because primary emission should not change during this period, the unfavorable meteorological conditions could be responsible for the event. Low WS and high RH (from 80% to >90%) was found from February 5–14, with a few hours of RH <60% during the daytime. Particle hygroscopic growth under high ambient RH conditions and heterogeneous reactions on particle surfaces could also contribute to the elevated particle mass concentration (X. Wang et al., 2018). Consequently, nucleated particles accumulated because of enhanced oxidizing capacity and favorable meteorological conditions and caused severe aerosol pollution.

## 4. Conclusion

In this study, we presented changes in the NSD of particles and charged ions measured in January and February 2020 to reveal the influence of emission reduction on NPF events. Particles smaller than 100 nm were effectively reduced by ~40% because of suspended human activities during the Chinese New Year and the COVID-19 LCD. The accumulation mode particles were slightly higher during LCD, as several hazy days were associated with secondary aerosol formation. The frequency of NPF days slightly varied; however, $J_2$ and $GR$ were significantly higher. During LCD, $NO_x$ and $SO_2$ concentrations decreased as anthropogenic emissions were reduced. Higher $O_3$ and a lower condensation sink raised sulfuric acid concentration levels by ~26% , which were responsible for the higher nucleation rate and larger nanoparticle quantity. Sulfuric acid was also responsible for the nucleated particle growth at the initial sizes (below 5 nm). In the late afternoon,

sulfuric acid decreased as the weakend solar radiation and low volatility oxidation products of VOCs could have larger contribution to the particle growth. For the major VOCs, isoprene and toluene were slightly changed, benzene increased, and aromatics (C8 and C9 componds) decreased during LCD period. Although the oxidation products of VOCs were not measured

in this study, the major oxidants of VOCs ($O_3$, OH and $NO_3$) all increased during LCD period, indicating the possibility of enhanced HOMs promoting the particle growth process. The effect of charge ions on the particle growth was also studies and it showed a enhanced growth rate facotor of 1.38±0.34. The nucleated particles entered accumulation mode by the secondary aerosol formation and underwent hygroscopic growth under high RH and calm wind conditions, which facilitated the occurrence of severe pollution episodes during LCD. This work highlights the potential influence of strict emission control

strategies on NPF events and provides insights into the positive and negative effects of precursors and atmospheric oxidizing capacity on the nucleation and growth process of the nanoparticles.

*Data availability.* All the data related to this paper may be requested from the corresponding author: shenxj@cma.gov.cn.

*Author contributions.* XS and JS designed the research and led the overall scientific questions. XS, JZ, YZ, YW, CX, XH and SZ carried out the field experiment, data processing and analysis. XS wrote the first draft of the manuscript and FY, XZ revised

the manuscript. All authors read and approved the final version.

*Competing interests.* The authors declare that they have no conflict of interest.

*Acknowledgement.* This research was supported by the National Key R&D Program of China (grant no. 2018YFC0213204), the National Natural Science Foundation of China (41875147, 42075082, 41675129) and the Chinese Academy of Meteorological Sciences (2020KJ001). It was also supported by the Innovation Team for Haze-fog Observation and Forecasts

of MOST and CMA.

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

Table 1. Parameters characterizing NPF events during Pre_LCD, LCD, and Post_LCD, including NPF frequency,

formation rate ($J_{2,par}$, $J_{2,pos\_ion}$, $J_{2,neg\_ion}$) and growth rate ($GR_{2,par}$, $GR_{pos\_ion}$, $GR_{neg\_ion}$) of the total particles and charge ions, as well as condensation sink ($CS$).

| | Pre_LCD | LCD | Post_LCD |
|---|---|---|---|
| NPF event (available days) | 10 (23) | 10 (24) | 5 (13) |
| $J_{2,par}$ (cm$^{-3}$s$^{-1}$) | $5.6 \pm 2.3$ | $7.9 \pm 4.5$ | $5.9 \pm 3.5$ |
| $J_{2,pos\_ion}$ (cm$^{-3}$s$^{-1}$) | $0.010 \pm 0.003$ | $0.032 \pm 0.003$ | $0.021 \pm 0.014$ |
| $J_{2,neg\_ion}$ (cm$^{-3}$s$^{-1}$) | $0.009 \pm 0.004$ | $0.024 \pm 0.005$ | $0.015 \pm 0.011$ |
| $GR_{par}$ (nm h$^{-1}$) | $0.8 \pm 0.5$ | $1.5 \pm 0.7$ | $2.0 \pm 0.5$ |
| $GR_{pos\_ion}$ (nm h$^{-1}$) | $1.8 \pm 0.3$ | $3.1 \pm 0.2$ | $3.6 \pm 0.4$ |
| $GR_{neg\_ion}$ (nm h$^{-1}$) | $2.0 \pm 0.7$ | $3.1 \pm 0.4$ | $3.2 \pm 0.4$ |
| $CS$ (s$^{-1}$) | $0.010 \pm 0.003$ | $0.008 \pm 0.006$ | $0.008 \pm 0.003$ |

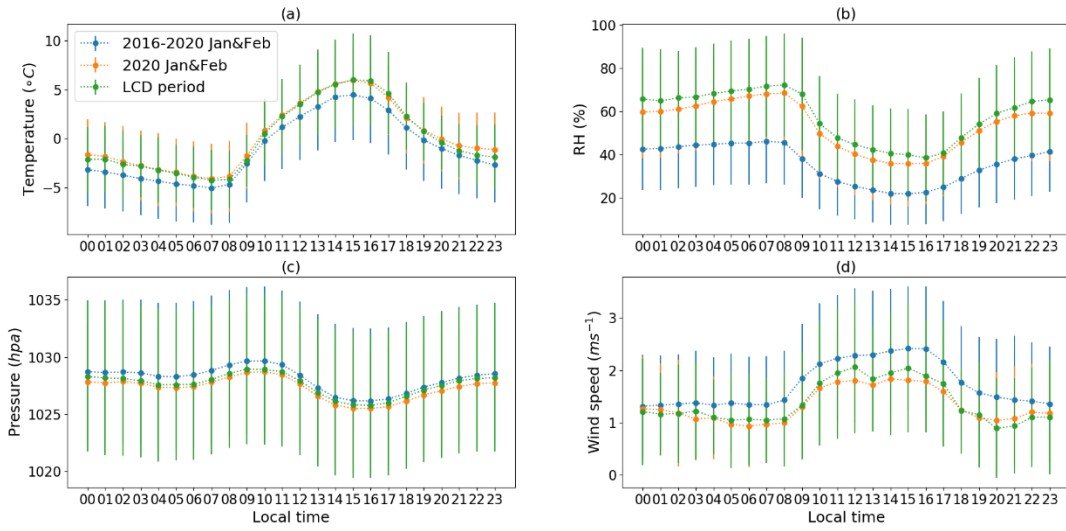

Fig. 1. The mean diurnal pattern of meteorological parameters, including temperature (a), RH (b), sea level pressure (c) and

 wind speed (d) during LCD period (January 24-February 16, 2020), January and February in 2020 and in 2016-2020. The

solid circles and bars represent the mean value and the standard deviation, respectively.

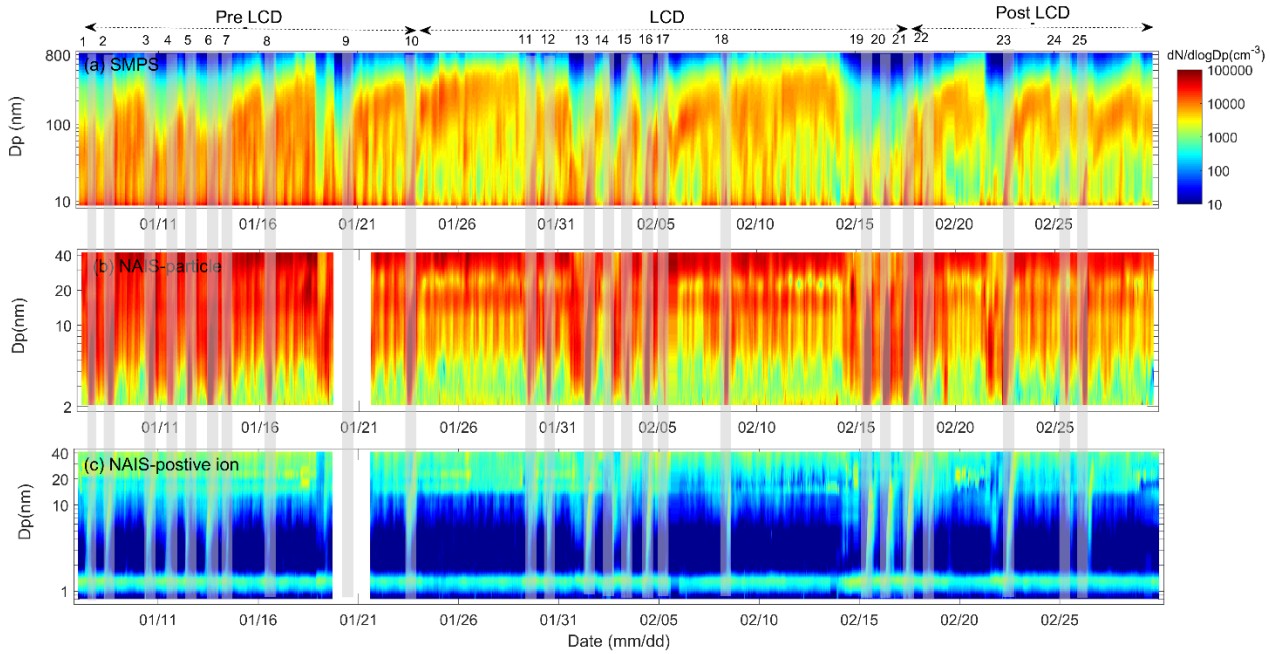

Fig. 2. Time evolution of number size distribution of 10–850 nm particles by SMPS (a), neutral 2–42 nm particles by NAIS

in positive particle mode (b), and positive 0.8–42 nm ions by NAIS (c). NPF events were marked by numbers from 1–25.

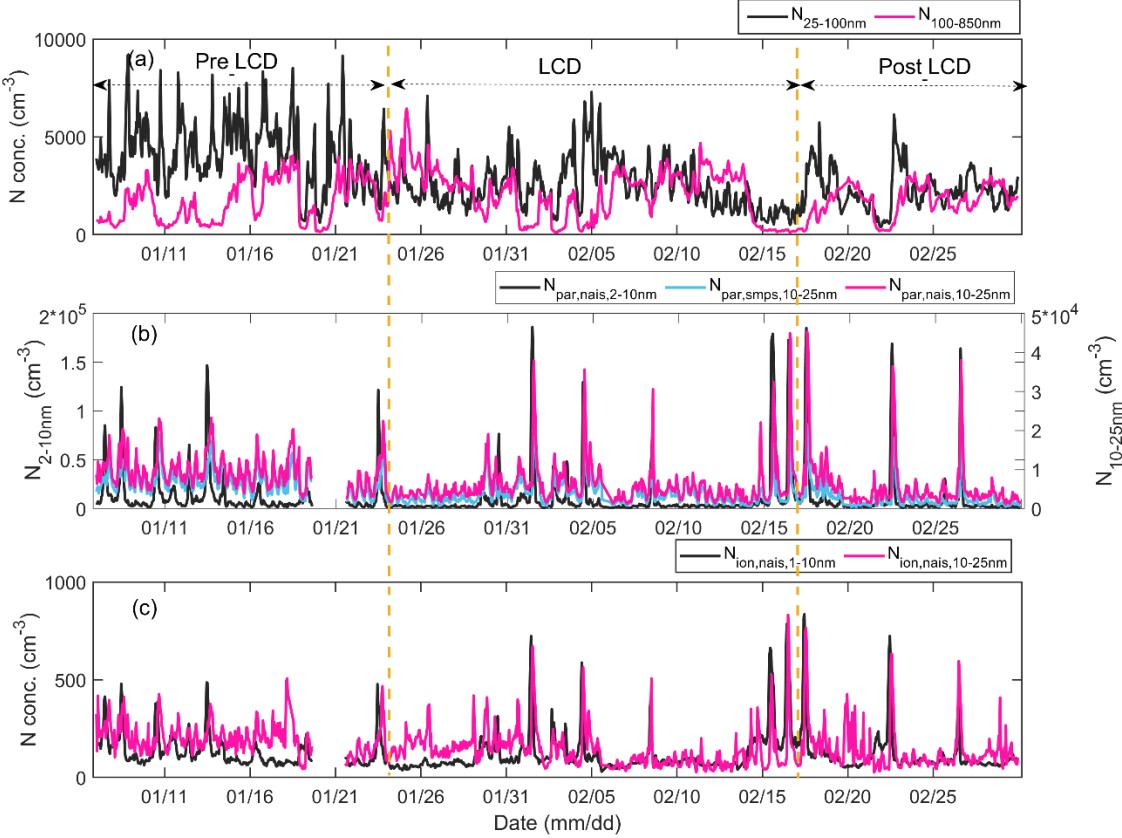

Fig. 3. Time series of hourly mean of the number concentrations for different size ranges, including 25–100 nm and

100–850 nm, from SMPS data (a); 2–10 nm particles from NAIS and 10–25 nm particles from both NAIS and SMPS

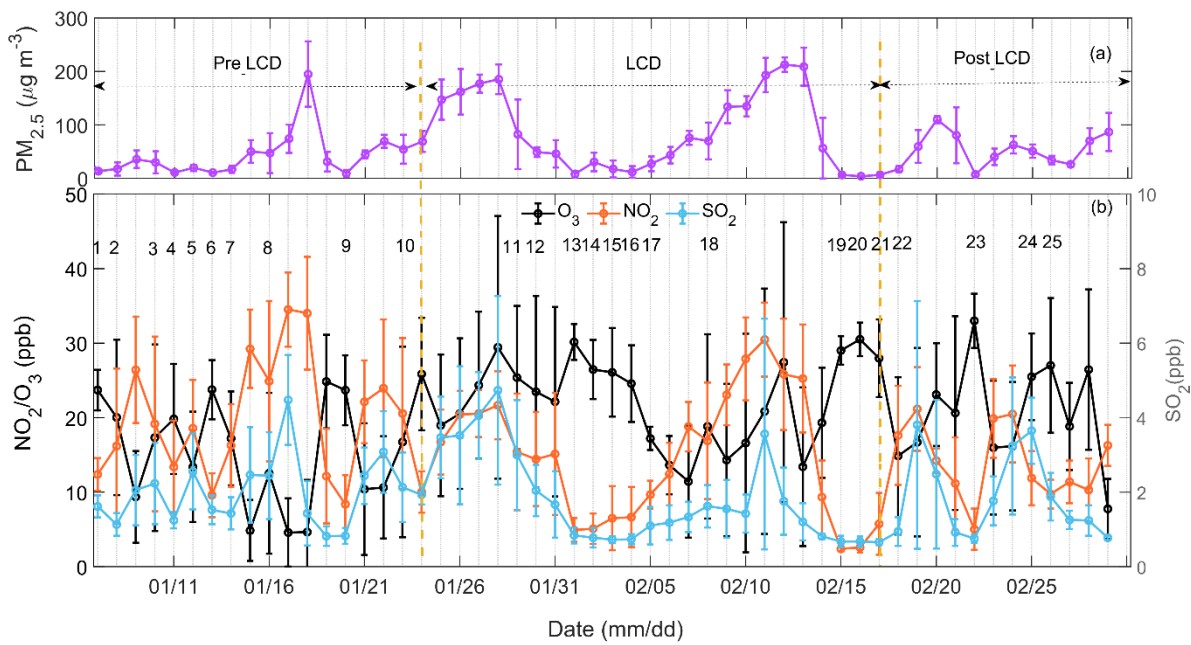

Fig. 4. Concentration level of PM$_{2.5}$ mass concentration (a), and precursors (b), including NO$_2$, SO$_2$, and O$_3$ during the measurement period. The circle and bar indicate the mean and standard deviation, respectively; NPF days are marked with continuous numbers 1-25.

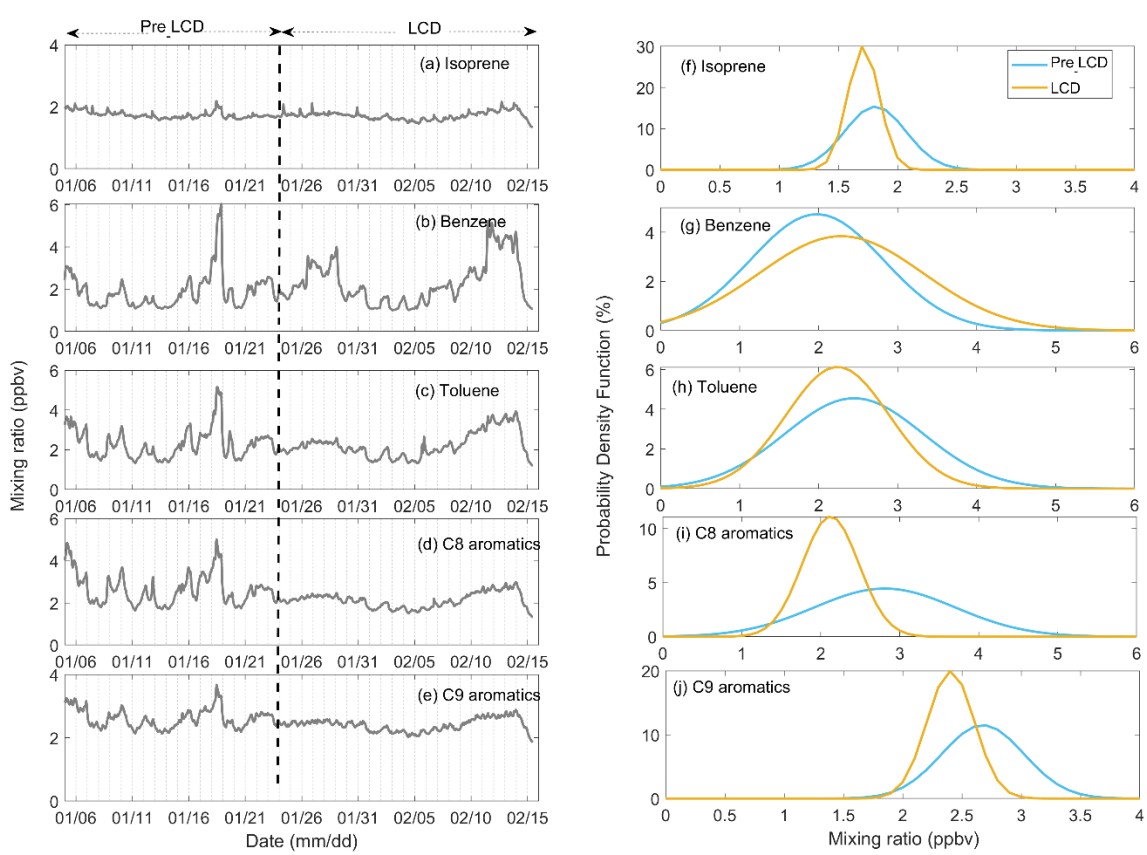

Fig. 5. Time series of isoprene, benzene, toluene, C8 and C9 aromatics (a-e) during January 5 to February 15, and the probability distribution function of mixing ration of each VOC component (f-j), respectively.

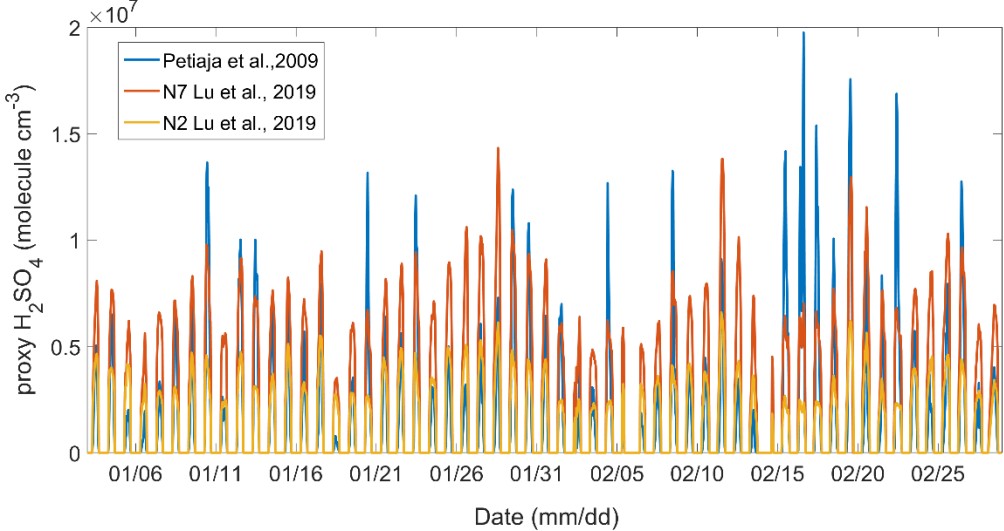

Fig. 6. The sulfuric acid concentrations derived by different proxy equations. The red and orange lines indicate the result by

N2 and N7 method by Lu et al. (2019), and blue line indicates the method recommend by Petiäjä et al. (2009).

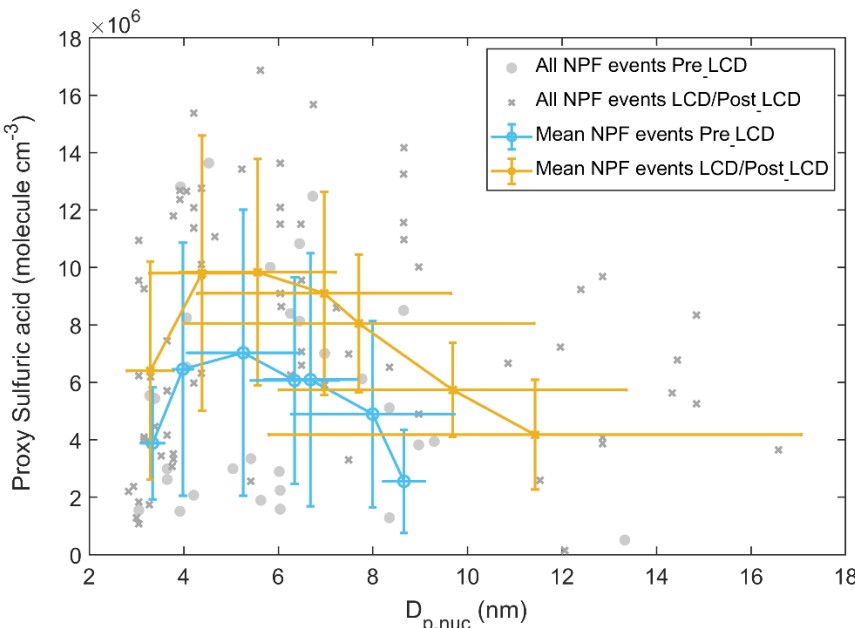

505

Fig. 7. Scatter plot between geometric mean diameter of nucleation mode ($D_{p,nuc}$) and the proxy sulfuric acid. The grey dots

and crosses represent the NPF events during Pre_LCD, LCD/Post_LCD, respectively. The purple and blue lines represent the

mean conditions during Pre_LCD, LCD/Post_LCD. The vertical and horizontal bars represents the standard deviations of

sulfuric acid and $D_{p,nuc}$.

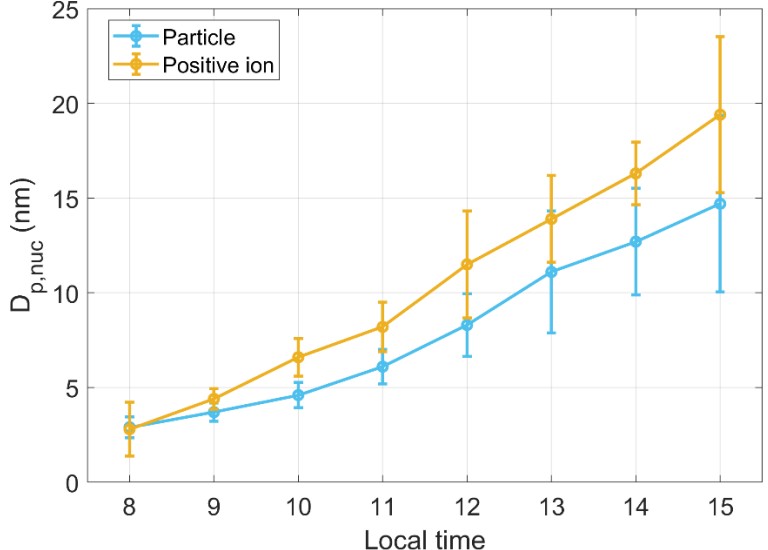

510

Fig. 8. The time evolution of geometric mean diameter of nucleation mode ($D_{p,nuc}$) of neutral particle and positive charged

ions during the NPF events. The circle and bar present the mean value and the standard deviation.

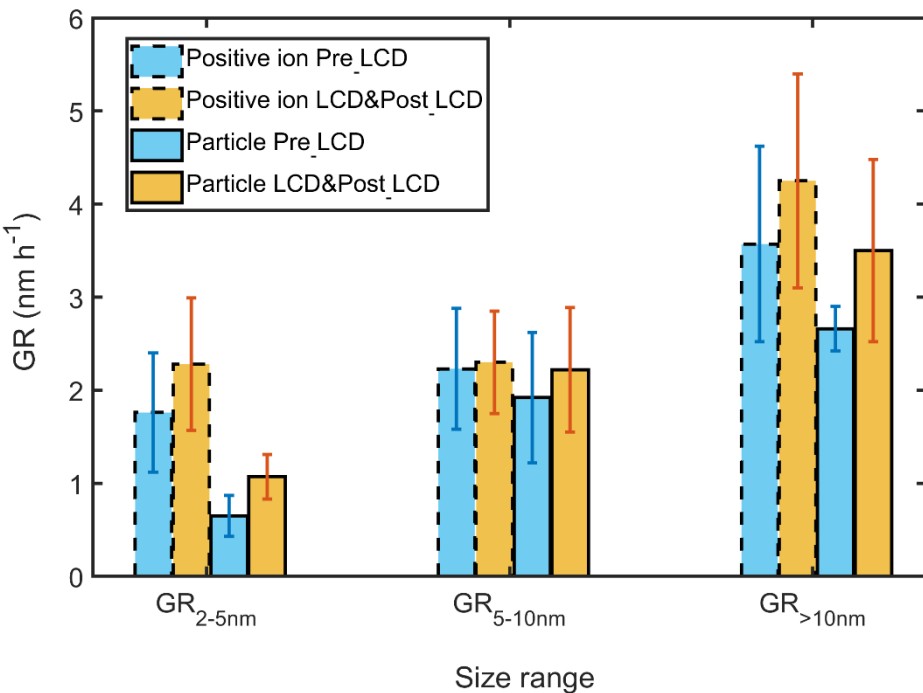

Fig. 9. The mean values of growth rates of particles and ions in different size ranges, including 2–5 nm ($GR_{2-5nm}$), 5–10

515      nm ($GR_{5-10nm}$), and >10 nm ($GR_{>10nm}$), during Pre_LCD and LCD&Post_LCD, respectively. The histogram and error

bars represent the mean value and standard deviation, respectively.

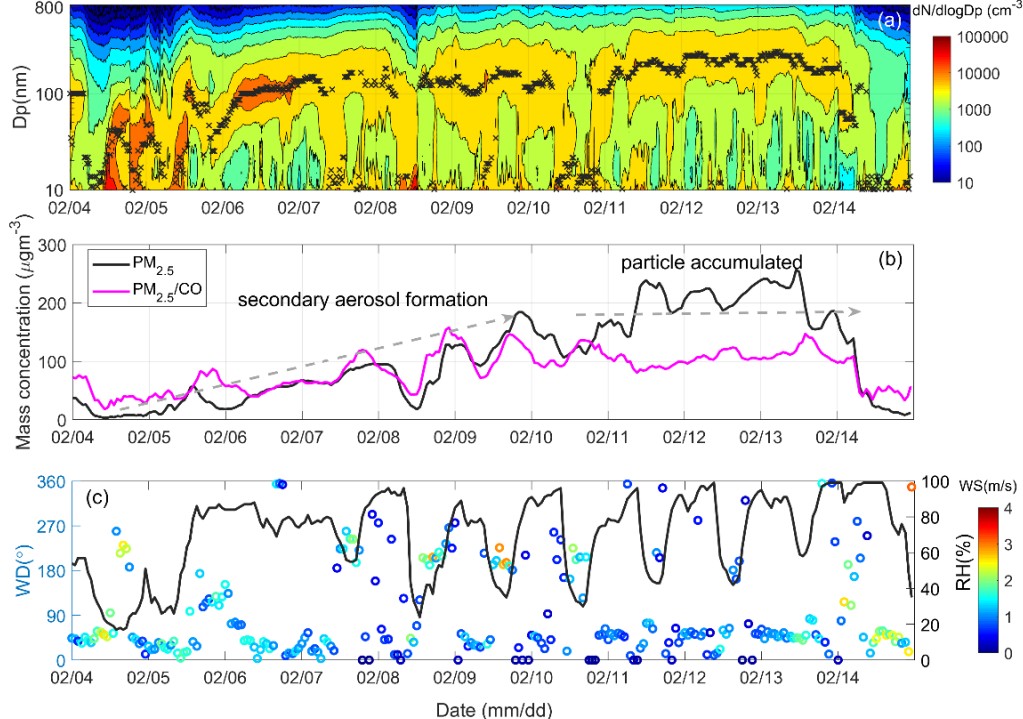

Fig. 10. Time evolution of PNSD and the dominant geometric mean diameters (black cross) derived by the log-modal fitting (a) and hourly mean PM$_{2.5}$ (black line), normalized PM$_{2.5}$ by CO (pink line) in (b) and the meteorological factors: wind direction (WD), wind speed (WS), and relative humidity (RH, black line) in (c) on February 4–14.

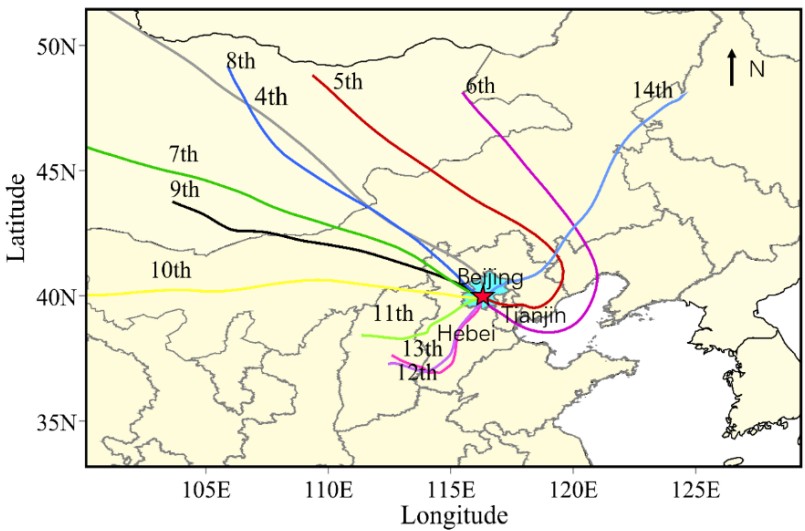

Fig. 11. The back trajectories arriving at CAMS station at 12:00 local time form February 4 to 14 with the terminal height of 500 m agl.