# Peer review of "Enhancement of nanoparticle formation and growth during the COVID-19 lockdown period in urban Beijing"

_Atmospheric Chemistry and Physics, 2020_

## Referee Comment (RC1) · Anonymous Referee #1 · 1 Dec 2020

In this study, the authors compare observations before, during and after the lockdown period in China during January and February 2020. They observe an enhancement of the nucleation and growth process of nanoparticles during the lockdown in Beijing. From this, they conclude that these findings were caused by the lockdown period, mainly due to lower concentration of Aitken mode aerosols that reduces the condensation sink. In contrast, accumulation mode particles increased and caused pollution events, due to new particle formation events with subsequent growth. This enhanced particle nucleation and growth is attributed to enhanced values of H2SO4 and VOC oxidation products, which were calculated from available data.

[Figure]

The subject of this study is certainly suited for ACP.

However, I have two major concerns regarding this study that need to be addressed in a revised version. These concerns are a) meteorological representativeness and b) statistical significance. I explain my concerns in detail in the following.

Major issues:

a) Meteorological representativeness

A major problem when comparing air pollution data from different periods is the influence of meteorology. This needs to be considered to ensure that the observed differences are not coincidence. High and low pressure systems may prevail for a certain time, leading to differences in cloud coverage and thereby to enhanced or reduced radiation. Wind speed and direction influences transport of pollution from sources, either regional or even from long distances. Meteorological parameters are available. I suggest conducting something like a 5-year climatology of the available parameters to check the variability of the atmosphere and the representativeness of the Jan-Feb 2020 period, especially the lockdown period, compared to the same period in previous years. In line 40-41, the authors state that "Furthermore, particle accumulation was favored by stagnant airflow and vertical meteorological conditions during LCD (Zhong et al., 2020)." So apparently, they are aware of unusual meteorological conditions during the LCD period. But, since the Zhong et al. reference is still in preparation, the reader can't retrieve this additional but important information.

b) Statistic significance

The changes in NPF event frequency seem not to be significant, because the number of NPF events is small. The authors report on differences of "10 out of 23 days (43%)", "8 out of 24 days (33%)", and "5 out of 13 days (38%)". These are small numbers. A simple estimation based on Poisson statistics suggests that these differences may not be significant. Here a detailed statistical analysis has to be presented, and it may be

that the results will be that these differences are not significant.

Similar analyses have to be done for pollutants NO2 and SO2, because Fig. 3 shows that both are highly variable during the pre-LCD, LCD, and post-LCD periods.

The linear regressions between H2SO4, J2, and GR are significant (Fig 5), but that's not new. And since H2SO4 is mainly calculated from global radiation, the meteorological influence on this parameter is high. The different growth rates when comparing pre-LCD, LCD and post-LCD for the different size ranges presented in Fig 6 may also not be significant, regarding the error bars.

Minor comments

Line 20: Please explain the meaning of J2 also in the Abstract.

Line 34-35: Reformulate this sentence to: The number of Aitken mode particles (d $\sim 25 - 100$ nm), which is related to traffic emissions (Deventer et al., 2018) is also expected to decrease.

Line 38: Change "Air pollution is driven by the enhancement of secondary particles," to "Secondary particles contribute significantly to air pollution"

Line 41: The reference to Zhong et al. in preparation is not sufficient. If you do not want to show these data now, then I suggest to wait until the Zhong et al. paper is submitted as well. The meteorological situation (e.g. the inversion layer) is of great importance (see my major comment above).

Line 107 – 113: H2SO4 estimation: This is a very rough estimation. What are usual values for k, and what is the dimension of k? In Figure 4 no units are given on the right y-axis, but shouldn't that be cm-3?

Lines 123-130: So, to infer VOC oxidation capacity, you don't have OH measurements, thus you approximate OH, but you don't have UVB, so you approximate by global radiation. This seems like many uncertainties. Can you comment a little more on the

uncertainties and the influence they might have on your interpretation?

Line 126, Equ. 6: There is a ratio of two numbers (8.4e-7/8.6e-10). Are these numbers rate coefficients that should have units? Or are they just empirical fit parameters? If they are just parameters, you can replace them by 9.8e2.

Line 132-135: What are the exact criteria to define the NPF events?

Line 140-142 & 178: As already commented above: "The NPF event occurred on 10 out of 23 days (43%) during pre-LCD, 8 out of 24 days (33%) in LCD, and 5 out of 13 days (38%) in post-LCD, respectively". The frequencies of 43% or 33% are based on a very low number of events. Please add the total number of events to the table, and calculate Binomial or Poisson statistics for these numbers. It may be that the differences are too small to be significant, just by the small number of events.

Line 150: Please refer to Fig 2a here and replace "...were discussed in detail." by "are discussed in detail in the following". Otherwise it is hard for the reader to follow this discussion.

Line 170/171: Refer to Fig 2b here.

Line 174: Refer to Fig 2c.

Line 180: What is J3? Should it read J2? Or does this refer to measurements from 2015 where 3 nm was the lower size? If so, please explain.

Line 182-184: "in this study", "The previous study". Please make clear which study is which.

Line 191: " As discussed separately for LCD and pre-LCD during the NPF event occurrence (9:00–16:00 LT),...". Which NPF event are you talking about? You specify the time period 9:00 – 16:00 LT. But Fig. 4 shows a time series of the whole campaign. Individual events can not be seen here. The whole paragraph line 191 until line 200 can not be understood, because you refer to one NPF event that is not shown. Which day

is that? I assume that this text describes an event that was discussed in Huang et al 2020, but if that is so, this discussion does not belong here in this paper. Please include a time series of the measured parameters for this event here or skip this paragraph.

Line 201: " . . .was indicated by different VOC_ox,capacity levels" seems not to be the right expression here. I suggest to rephrase the whole sentence: "Both the H2SO4 proxy and the VOX_ox,capacity were correlated to J_2,tot and to GR (Fig 5)."

Line 205-207: This is again a result from another study. Make that clear at the beginning of the sentence, like "Stolzenburg et al. (2020) showed that sulfuric acid could not explain. . .."

Lines 208-210: If H2SO4 contributed more to the nucleation process and organic vapours to the growth, would you not expect to see a better correlation between H2SO4 and J2 than between H2SO4 and GR, and similarly a better correlation between VOC_ox,cap and GR than between VOC_ox,cap and J2?

Lines 214-215: For the size range 5-10 nm, there is no significant difference between ions and neutral particles. Especially the yellow bars for 5-10 nm (Fig. 6) have almost exactly the same height. What are the error bars and what is their meaning? This should explained in the caption of Fig 6.

Line 220: What is the enhancement factor? How is it calculated?

Line 223: Replace "effect of the charger" by "effect of charge"

Lines 228-231: It would be helpful to add PM2.5 to one of the time series in Fig. 3, and to add the numbers of the NPF events in Fig 3 and 4 instead of or additional to the crosses. When I count the NPF events marked by the crosses, I find that event #9 is on January 23.

Lines 233-238: Please include a graph showing PM2.5/CO. Please also state clearly how pollution periods were identified.
* * *
**Interactive comment**

Line 238-240: This sentence doesn't make sense. Maybe the "and" before "unfavorable" needs to be deleted?

Whole Section 3.4: What is the message of the section 3.4? The reader can not see the pollution events if there is no graph showing PM2.5, or CO, or both. Are there any conclusion drawn by section 3.4? It all seems very speculative. Meteorological conditions are mentioned as one possible reason for this pollution event, but it is not investigated by trajectory and emission source locations.
* * *

---

## Referee Comment (RC2) · Anonymous Referee #2 · 31 Dec 2020

This study presents observational results of particle and ions number concentration in Beijing, China during Jan. 24 - Feb. 14, 2020. This period represent the Chinese New year and the COVID-19 lockdown period, therefore suitable for understanding the influence of reduced emission to nanoparticle formation and growth. Although the data set is novel and the main research question it wants to discuss sounds interesting, there are too many important values and discussion in the text which are based on guessing. Overall, I don't recommend this paper to be considered to be published before major reconstruction.

1.I would say that the current evidence (No measured or modelled VOCs) does not

allow the major conclusion on VOC contribution of particle growth of particles of 10nm~100nm. The only mentioning of VOCs is that reduces by ~45% in the BTH area, but then they say that low volatility products increased with a VOCs oxidation capacity factor of ~1.3. But if we do a simple estimation, 1.3*0.55=0.71, which would mean the oxidation products would not increase. I'm not questioning the conclusion that oxidation products could increased, but there's just far too less data in this study to support this.

2.Furthermore, it seems to me at least that authors does not understand the concept of ELVOCs or HOMs correctly. It should be noted that the concept of "ELVOCs" should be used when we don't have information of the volatility, but only use "HOMS" when we are discussing oxidized VOCs. However, using k1*[OH]*[VOC]+k2*[OH]*[VOC] is certainly not acceptable as a proxy of HOMs. First, this represents the first generation product of oxidized VOCs, and even for a-pinene oxidation which is the most studied HOMs production pathway, this only produces very volatile OVOCs, and they won't contribute substantially for the growth of Aitken mode particles. Secondly, while this proxy was developed for an a-pinene rich boreal forest, the main VOCs in Beijing are aromatic, alkenes, and even the main BVOC are not a monoterpene but isoprene. So there needs to be far more discussion to settle down which is the main OVOC contributing as low volatile products, and multi-generation products instead of first-order products should be considered. Last but not least, in the wintertime in Beijing, the night is very long, and oxidation by NO3 should not be ignored.

3.The authors claim that nucleated particles can grow to CCN size and contribute to particle mass during haze events. But overall, there is very less discussion of investigation of the particle growing form nucleation mode to accumulation mode, which means that quantitative understanding is lacking. For Fig7., if we take 6th Feb for example, it seems that the growth from aitken to accumulation mode comes from growth of pre-existing particles. And even though the growth of particle number concentration seems to terminate by 7th Feb, it seems that the PM2.5 mass still increases rapidly. At

noon 8th Feb, it looks like there is a new polluted air mass coming, leading for stronger pollution.

4.I think the paper can be resubmitted by putting more effort on nucleation and early growth by sulfuric acid. The NAIS measurement seems to work good, and could be discussed more in depth. To explain the growth driven by oxidized VOCs, either support by measurement of chemical composition or a chemical mechanism model is needed.

Minor comments:

1.The fitting coefficients for H2SO4 proxy should not used the same as the measurement in Finland. For Beijing, there's a paper by Lu et al (2019). Note that the effect is nonlinear and will effect trend in Fig 4-5. And if Global Radiation is used instead of UVB for H2SO4, it should be stated as it was done for OH. UVB is a fraction of Global Radiation, therefore a new coefficient should be used. Lu, Y., Yan, C., Fu, Y., Chen, Y., Liu, Y., Yang, G., Wang, Y., Bianchi, F., Chu, B., Zhou, Y., Yin, R., Baalbaki, R., Garmash, O., Deng, C., Wang, W., Liu, Y., Petäjä, T., Kerminen, V.-M., Jiang, J., Kulmala, M., and Wang, L.: A proxy for atmospheric daytime gaseous sulfuric acid concentration in urban Beijing, Atmos. Chem. Phys., 19, 1971–1983, https://doi.org/10.5194/acp-19-1971-2019, 2019.

2.I didn't found the OH proxy used here in Petäjä et al(2009). Please make sure the right reference is cited. 3.There are spelling mistakes and grammar errors, try to find an english expert to fix them all, eg: The number concentration of Aitken mode particles (∼25-100nm) should also decreased as expected-> The number concentration of Aitken mode particles (∼25-100nm) decreased as expected or The number concentration of Aitken mode particles (∼25-100nm) decreased as they should.

Use the term oxidizing capacity consistently, replace all "oxidization capacity"

---

## Author Comment (AC1) · 20 Feb 2021

Response to RC1

In this study, the authors compare observations before, during and after the lockdown period in China during January and February 2020. They observe an enhancement of the nucleation and growth process of nanoparticles during the lockdown in Beijing. From this, they conclude that these findings were caused by the lockdown period, mainly due to lower concentration of Aitken mode aerosols that reduces the condensation sink. In contrast, accumulation mode particles increased and caused pollution events, due to new particle formation events with subsequent growth. This enhanced

particle nucleation and growth is attributed to enhanced values of H2SO4 and VOC oxidation products, which were calculated from available data. The subject of this study is certainly suited for ACP. However, I have two major concerns regarding this study that need to be addressed in a revised version. These concerns are a) meteorological representativeness and b) statistical significance. I explain my concerns in detail in the following.

Response: The authors thank the reviewer's comments and try our best to address the issues point-by point.

Major issues:

a) Meteorological representativeness A major problem when comparing air pollution data from different periods is the influence of meteorology. This needs to be considered to ensure that the observed differences are not coincidence. High and low pressure systems may prevail for a certain time, leading to differences in cloud coverage and thereby to enhanced or reduced radiation. Wind speed and direction influences transport of pollution from sources, either regional or even from long distances. Meteorological parameters are available. I suggest conducting something like a 5-year climatology of the available parameters to check the variability of the atmosphere and the representativeness of the Jan-Feb 2020 period, especially the lockdown period, compared to the same period in previous years. In line 40-41, the authors state that "Furthermore, particle accumulation was favored by stagnant airflow and vertical meteorological conditions during LCD (Zhong et al., 2020)." So apparently, they are aware of unusual meteorological conditions during the LCD period. But, since the Zhong et al. reference is still in preparation, the reader can't retrieve this additional but important information.

Response: Thanks for the reviewer's constructive comment. (1) The meteorological parameters during LCD period, January and February in 2020, as well as the average conditions of January and February in 2016-2020 were analyzed and the diurnal pat-

ten was given (Fig. 1). It showed much higher RH, lower wind speed, slightly higher temperature and lower pressure during LCD and January and February 2020, than that of 5-year climatology average condition (January and February in 2016-2020). The anomaly of monthly mean sea level pressure in January and February between 2020 and 2016-2020 was analyzed based on the ECMWF reanalysis dataset (ERA5, https://cds.climate.copernicus.eu/), as given in Fig. 2. It showed negative anomaly in Beijing-Tianjin-Hebei Province, indicating the air pressure decreased in January and February in 2020, as compared with the corresponding period of the 5-year climatology. The local air convergence resulted in high RH and low wind speed, which favored for the air pollutants accumulating. (2) In line 40-41, we removed the Zhong et al., 2020 in the manuscript and referred the previous publication (Zhong et al., 2018) also conducted in Beijing to address the meteorological effect on the air pollution formation. The discussion was supplemented in the section of "3.1 The meteorological conditions".

b) Statistic significance The changes in NPF event frequency seem not to be significant, because the number of NPF events is small. The authors report on differences of "10 out of 23 days (43%)", "8 out of 24 days (33%)", and "5 out of 13 days (38%)". These are small numbers. A simple estimation based on Poisson statistics suggests that these differences may not be significant. Here a detailed statistical analysis has to be presented, and it may be that the results will be that these differences are not significant. Similar analyses have to be done for pollutants $NO_2$ and $SO_2$, because Fig. 3 shows that both are highly variable during the pre-LCD, LCD, and post-LCD periods. The linear regressions between $H_2SO_4$, J2, and GR are significant (Fig 5), but that's not new. And since $H_2SO_4$ is mainly calculated from global radiation, the meteorological influence on this parameter is high. The different growth rates when comparing pre-LCD, LCD and post-LCD for the different size ranges presented in Fig 6 may also not be significant, regarding the error bars.

Response: (1) The Poisson statistics was conducted for NPF event occurrence probability for pre-LCD, LCD and post-LCD period, respectively, as given in Fig. 3. It showed

almost the same NPF event occurrence probability as compared with pre-LCD and LCD period, but fewer NPF event during Post_LCD. (2) The probability density function (PDF) was given for SO2, NO2 and O3 during Pre_LCD, LCD and Post_LCD, respectively, as given in Fig. 4. It showed significant decreasing trend of NO2, whereas increasing trend of O3 as compared with Pre_LCD and LCD/Post_LCD. However, the variation of SO2 among different periods was not clear, as the SO2 concentration remained low due to the emission control these years. The PDF of gases pollutants and the detailed discussion have been added in the text and the figure is given in the supplementary materials. (3) H2SO4 proxy was derived by three methods and the average value was applied for discussion, in order to minimize the uncertainties. (4) the growth rate (GR) was discussed for different size range 2-5 nm, 5-10 nm and >10 nm, respectively. It showed GR2-5nm and GR>10nm were generally higher during LCD/Post_LCD as compared with Pre_LCD, however, the difference of GR5-10 nm is not clear.

Minor comments

Line 20: Please explain the meaning of J2 also in the Abstract.

Response: It has been revised to be "higher formation rate of 2 nm particles (J2) and the subsequent growth rate (GR)".

Line 34-35: Reformulate this sentence to: The number of Aitken mode particles (d∼ 25 – 100 nm), which is related to traffic emissions (Deventer et al., 2018) is also expected to decrease.

Response: It has been revised in the text.

Line 38: Change "Air pollution is driven by the enhancement of secondary particles, " to "Secondary particles contribute significantly to air pollution"

Response: It has been revised in the text.

Line 41: The reference to Zhong et al. in preparation is not sufficient. If you do not want to show these data now, then I suggest to wait until the Zhong et al. paper is submitted

as well. The meteorological situation (e.g. the inversion layer) is of great importance (see my major comment above).

Response: we removed the Zhong et al., 2020 in the manuscript and referred the previous publication (Zhong et al., 2018) also conducted in Beijing to address the meteorological effect on the air pollution formation. The discussion was supplemented in the section of "3.1 The meteorological conditions". Zhong, J., Zhang, X., Dong, Y., Wang, Y., Liu, C., Wang, J., Zhang, Y. and Che, H.: Feedback effects of boundary-layer meteorological factors on cumulative explosive growth of PM2.5 during winter heavy pollution episodes in Beijing from 2013 to 2016, Atmospheric Chemistry and Physics, 18(1): 247-258, DOI: 10.5194/acp-18-247-2018, 2018.

Line 107 – 113: H2SO4 estimation: This is a very rough estimation. What are usual values for k, and what is the dimension of k? In Figure 4 no units are given on the right y-axis, but shouldn't that be cm-3?

Response: The concentration of H2SO4 was not measured directly in this study and different proxy methods were referred to derive the proxy sulfuric acid. A method (Eq. 3) depends on the global radiation (Glob_R), SO2 and condensation sink (CS), and is developed according to the previous study conducted in a forest site, Hyytiälä, Finland (Petäjä et al., 2009). [H2SO4]=(k×Glob_R×SO2)/CS (1) where k is empirically derived factor and well correlated with Glob_R (k=1.4*10-7*Glob_R-0.7, unit: m2 W-1 s-1). The proxy equation is site-specific due to the different atmospheric conditions. In the polluted atmosphere, such as in Beijing, several proxy methods were also constructed based on a number of available atmospheric parameters (Lu et al., 2019). In this study, the simplest proxy (Eq. 4) and best performance proxy (Eq. 5) are adopted to derive the proxy sulfuric acid. [H2SO4]=280.05×UVBˆ0.14×[SO2 ]ˆ0.40 (2) [H2SO4]=0.0013×UVBˆ0.13×[SO2 ]ˆ0.40×CSˆ(-0.17)×([O3 ]ˆ0.44+[NOx ]ˆ0.41) (3) [H2SO4] is the gaseous sulfuric acid with the unit of molecule cm-3. [SO2], [O3] and [NOx] is the concentration of sulfur dioxide, ozone, and nitrogen oxides, with the unit of molecule cm-3. UVB is the intensity of ultraviolet radiation b in W m-2. CS is

the condensation sink, which describes how fast the vapor molecules condense on the existing particles (Dal Maso et al., 2002), with the unit of s-1. The proxy method has been validated by comparing the measured sulfuric acid with a high correlation coefficient of 0.86 (Lu et al., 2019), based on the field campaign conducted approximately 2 km away from CAMS site. In this work, the direct measurement of UVB was not available. However, it had been reported by Hu et al. (2013) that the monthly average of the ratio of UVB to global radiation (Glob_R) ranged from 0.007 to 0.017% in Beijing. And in this study, the average ratio of January and February (0.008%) was applied to derive UVB by 0.008%*Glob_R.

Lines 123-130: So, to infer VOC oxidation capacity, you don't have OH measurements, thus you approximate OH, but you don't have UVB, so you approximate by global radiation. This seems like many uncertainties. Can you comment a little more on the uncertainties and the influence they might have on your interpretation?

Response: As discussed above, the UVB was derived by 0.008% * Glob_R, based on the previous study that the monthly average of the ratio of UVB to global radiation (Glob_R) ranged from 0.007 to 0.017% in Beijing (Hu et al., 2013). The average ratio of January and Feburary (0.008%) was applied in this study for calculating [H2SO4].

Line 126, Equ. 6: There is a ratio of two numbers (8.4e-7/8.6e-10). Are these numbers rate coefficients that should have units? Or are they just empirical fit parameters? If they are just parameters, you can replace them by 9.8e2.

Response: The equation has been removed from the manuscript.

Line 132-135: What are the exact criteria to define the NPF events?

Response: It has been revised to "NPF events are identified and different nucleation types are characterized based on the daily evolution of particle number size distribution (PNSD). The burst of nucleation mode particles with diameter $\leq$ 25 nm appeared in the PNSD, and the burst should prevail over a few hours with clear growth process (Dal

Maso et al., 2005)."

Line 140-142 & 178: As already commented above: "The NPF event occurred on 10 out of 23 days (43%) during pre-LCD, 8 out of 24 days (33%) in LCD, and 5 out of 13 days (38%) in post-LCD, respectively". The frequencies of 43% or 33% are based on a very low number of events. Please add the total number of events to the table, and calculate Binomial or Poisson statistics for these numbers. It may be that the differences are too small to be significant, just by the small number of events.

Response: We gave the number of NPF days and total available measurement days in table 1, instead of the NPF frequency. The Poisson statistics was conducted for NPF event occurrence probability for pre-LCD, LCD and post-LCD period, respectively, as given in Fig. S1. It showed fewer NPF events with higher probability as compared with pre-LCD and LCD period.

Line 150: Please refer to Fig 2a here and replace ". . .were discussed in detail." by "are discussed in detail in the following". Otherwise it is hard for the reader to follow this discussion.

Response: The sentence has been revised to "were given in Fig. 2a and discussed in detail in the following."

Line 170/171: Refer to Fig 2b here.

Response: It has been revised in the text.

Line 174: Refer to Fig 2c.

Response: It has been revised in the text.

Line 180: What is J3? Should it read J2? Or does this refer to measurements from 2015 where 3 nm was the lower size? If so, please explain.

Response: A sentence was given to explain the meaning of J3 in Line 185 "J3 referred the formation rate at 3 nm calculated from the particle number concentration of 3–4-

nm particles by Eq. (1), as the lowest detection limit of SMPS applied in 2015 and 2010-2013 campaign was 3 nm."

Line 182-184: "in this study", "The previous study". Please make clear which study is which.

Response: "in this study" refers to the results of this manuscript and "the previous study" are the references we cited (Lehtipalo et al., 2018, Yan et al., 2020). The sentences have been revised to be "The daily mean value of NO2 decreased by ∼35% and SO2 decreased by ∼13%, whereas O3 increased by 80% during LCD as compared to pre-LCD in this work (Fig. 3). Previous studies had indicated that NOx suppressed NPF events by influencing the formation of highly oxygenated organic molecules (HOMs), which participated in nucleation and initial particle growth (Lehtipalo et al., 2018; Yan et al., 2016; 2020)."

Line 191: " As discussed separately for LCD and pre-LCD during the NPF event occurrence (9:00–16:00 LT),...". Which NPF event are you talking about? You specify the time period 9:00 – 16:00 LT. But Fig. 4 shows a time series of the whole campaign. Individual events can not be seen here. The whole paragraph line 191 until line 200 can not be understood, because you refer to one NPF event that is not shown. Which day is that? I assume that this text describes an event that was discussed in Huang et al 2020, but if that is so, this discussion does not belong here in this paper. Please include a time series of the measured parameters for this event here or skip this paragraph.

Response: "the NPF event occurrence (9:00–16:00 LT)" indicates all the NPF events usually occur during the daytime (9:00-16:00 LT), not refers to a specific NPF event. And the paragraph line 191 until line 200 are the discussion based on Fig. 4, describing the general characteristics of all NPF events during the measurement. We reorganized this paragraph to make it clearer.

Line 201: " : : :was indicated by different VOC_ox,capacity levels" seems not to be the right expression here. I suggest to rephrase the whole sentence: "Both the H2SO4

proxy and the VOX_ox,capacity were correlated to J_2,tot and to GR (Fig 5)."

Response: The sentence has been revised according to reviewer's comment.

Line 205-207: This is again a result from another study. Make that clear at the beginning of the sentence, like "Stolzenburg et al. (2020) showed that sulfuric acid could not explain: : :."

Response: the sentence has been revised to "Stolzenburg et al. (2020) revealed that sulfuric acid played an important role in smaller growth processes from 2–10 nm, however, could not explain condensational growth when the nucleated particles overcame 10 nm"

Lines 208-210: If H2SO4 contributed more to the nucleation process and organic vapours to the growth, would you not expect to see a better correlation between H2SO4 and J2 than between H2SO4 and GR, and similarly a better correlation between VOC_ox,cap and GR than between VOC_ox,cap and J2?

Response: The proxy sulfuric acid was re-calculated based on the reviewer's comment, and the effect of sulfuric acid on formation rate (J2) and initial growth rate (GR). The influence of H2SO4 on J2 and GR was re-evaluated, and it showed a slightly higher correlation coefficient (R) of J2 (R=0.62) than the GR (R=0.45). However, the oxidation product could not be estimated simply, but also discussed in the manuscript. The estimation of VOCs oxidizing capacity was removed in the manuscript, as the proxy method was not reasonable as the reviewer suggested. However, the direct measurement data of 5 major kinds of VOCs is supplemented and the variation is discussed. The major oxidants of VOCs were also found to be elevated during LCD, indicating the possibility of enhanced oxidation products of VOCs that promoted the nucleation and growth process.

Lines 214-215: For the size range 5-10 nm, there is no significant difference between ions and neutral particles. Especially the yellow bars for 5-10 nm (Fig. 6) have almost

exactly the same height. What are the error bars and what is their meaning? This should explained in the caption of Fig 6.

Response: The detailed growth process of the nucleated particles and ions on NPF days was given in Fig. 5. It showed Dp,nuc,ion grow faster than Dp,nuc,par, especially for the sizes below 5 nm, depending on the growth rate in each time interval ((Dp,nuc,t1- Dp,nuc,t2)/$\Delta$t, $\Delta$t = 1 h). The enhanced growth rate factor (GRp,nuc,ion/GRp,nuc,par) ranged from 1.1 to 2.0, with the average of 1.38+-0.34. The enhancement was higher in sub-10 nm particles, whereas it decreased as the particles grew to larger sizes. In addition, the histogram and error bars represent the mean value and standard deviation, respectively, which has been clarified in Figure caption.

Line 220: What is the enhancement factor? How is it calculated?

Response: The enhancement effect (EF) describes the dipole-charge interaction on the growth of charged clusters (Nadykto and Yu, 2003). The charge effect of ions on the NPF include accelerated rates of vapor condensation and particle coagulation, as well as the charge recombination (Yu and Turco, 1998; 2000). EF for the pure species participating NPF event (eg, sulfuric acid, VOCs) can be calculated as the below equation (Nadykto and Yu, 2003): EF=1+(2lE(r_p+r_m)L(lE(r_p+r_m )/kT+$\alpha\varepsilon$_0 E$\hat{}$2 (r_p+r_m)))/3kT (4) EF depends on temperature (T), size of charged particles (r_p), and microphysical properties (dipole moment: l, polarizability: ïĄą, and size: r_m) of vapor molecules. For sulfuric acid molecules, EF could be 10 for ions with ∼0.5 nm but decreased quickly to 2 for uptake by a charged particle of ∼2 nm, at T=300 K (Nadykto and Yu, 2003). In this work, we used enhancement factor (GRion/GRpar) to denote the net influence of charge on the particle growth. In order to differentiate with EF defined by equation 4, we used "enhanced growth rate factor" to denote GRion/GRpar. We also re-organized the section "effect of charge ions" to make it easier to understand. References: Nadykto, A. B. and Yu, F.: Uptake of neutral polar vapor molecules by charged clusters/particles: Enhancement due to dipole-charge interaction, J. Geophys. Res., 108(D23), DOI: 10.1029/2003jd003664, 2003. Yu, F. and Turco, R. P.: The formation and evolution of aerosols in stratospheric aircraft plumes: Numerical simulations and comparisons with observations, Journal of Geophysical Research: Atmospheres, 103(D20): 25915-25934, DOI: 10.1029/98jd02453, 1998. Yu, F. Q. and Turco, R. P.: Ultrafine aerosol formation via ion-mediated nucleation, Geophys. Res. Lett., 27(6): 883-886, DOI: 10.1029/1999gl011151, 2000.

Line 223: Replace "effect of the charger" by "effect of charge"

Response: It has been revised in the text.

Lines 228-231: It would be helpful to add PM2.5 to one of the time series in Fig. 3, and to add the numbers of the NPF events in Fig 3 and 4 instead of or additional to the crosses. When I count the NPF events marked by the crosses, I find that event #9 is on January 23.

Response: The figures has been revised in Fig. 6.

Lines 233-238: Please include a graph showing PM2.5/CO. Please also state clearly how pollution periods were identified.

Response: The discussion from lines 233-238 are referred to Fig. 7, which have contained PM2.5/CO in subplot (b). We revised the sentence as "Two principal pollution episode formation stages were identified according to variations in the PM2.5 mass concentration dividing by CO (PM2.5/CO), as indicated in Fig. 7b.". The pollution episodes were defined as the daily mean value of PM2.5 mass concentration exceeding 75 $\mu$g m-3, which is the criterion value of the second grade of air quality in China. We have added this sentence in the text.

Line 238-240: This sentence doesn't make sense. Maybe the "and" before "unfavorable" needs to be deleted?

Response: It has been revised in the text, "and" has been deleted.

Whole Section 3.4: What is the message of the section 3.4? The reader can not see

the pollution events if there is no graph showing PM2.5, or CO, or both. Are there any conclusion drawn by section 3.4? It all seems very speculative. Meteorological conditions are mentioned as one possible reason for this pollution event, but it is not investigated by trajectory and emission source locations.

Response: The discussion of section 3.4 is referred to Fig. 10 in the manuscript, including the evolution of PNSD, PM2.5 and its normalization by CO, which has been clarified in the text. The meteorological factor including wind direction, speed and relative humidity has been given in Fig. 10. Furthermore, the back-trajectory analysis from Feb, 4th-14th, corresponding to the study period in Fig. 10, is also supplemented (given as Fig. 11 in the manuscript). The back trajectories originated from northwest from February 4th to 10th, corresponding to the dry and clean air masses (Fig. 7). However, from February 11th to 13th, the southwesterly air masses were dominated and favored the accumulating of the particles, resulting in the high concentration level of particle matter. A paragraph of "2.5 Back trajectory analysis" is given in the section of "2. Method" In order to reveal the meteorological condition during the pollution case formation, the 48 h backward trajectories arriving at CAMS stie were calculated at 12:00 Local time, terminating at the height of 500 m above ground level by applying the Trajstat Software, combined with HYSPLIT 4 model (Hybrid Single-Particle Lagrangian Integrated Trajectory) and using the NCEP GDAS (Global Data Assimilation System) data with 1°ïĆť1° resolution (Draxler and Hess, 1998, Wang et al., 2009).
* * *
[Figure]

**Fig. 1.** The mean diurnal pattern of meteorological parameters, including temperature (a), RH (b), sea level pressure (c) and wind speed (d) during LCD period (January 24-February 16, 2020), January and Febru

[Figure]

**Fig. 2.** The anomaly of monthly mean sea level pressure in January and February be-
tween 2020 and 2016-2020. The data are from the ERA5 ECMWF reanalysis dataset
(https://cds.climate.copernicus.eu/).

[Figure]

**Fig. 3.** Poisson distribution of NPF event occurrence frequency during pre-LCD, LCD and post-LCD, respectively.

[Figure]

**Fig. 4.** The probability density function (PDF) of SO2, NO2 and O3 concentration during pre-LCD, LCD and post-LCD, respectively.

[Figure]

**Fig. 5.** The time evolution of geometric mean diameter of nucleation mode (Dp,nuc) of neutral particle and positive charged ions during the NPF events. The circle and bar present the mean value and th

[Figure]

**Fig. 6.** Concentration level of PM2.5 mass concentration (a), and precursors (b), including NO2, SO2, and O3 during the measurement period. The circle and bar indicate the mean and standard deviation, respecti

**Fig. 7.** The back-trajectory arriving at CAMS at 12:00 local time from Feb 4th to 14th, the star indicating the measurement site (CAMS) in Beijing.

---

## Author Comment (AC2) · 20 Feb 2021

This study presents observational results of particle and ions number concentration in Beijing, China during Jan. 24 - Feb. 14, 2020. This period represent the Chinese New year and the COVID-19 lockdown period, therefore suitable for understanding the influence of reduced emission to nanoparticle formation and growth. Although the data set is novel and the main research question it wants to discuss sounds interesting, there are too many important values and discussion in the text which are based on guessing. Overall, I don't recommend this paper to be considered to be published before major reconstruction.

Response: The authors thank the reviewer's comments and try our best to address the issues point-by point.

1. I would say that the current evidence (No measured or modelled VOCs) does not allow the major conclusion on VOC contribution of particle growth of particles of 10nm-100nm. The only mentioning of VOCs is that reduces by $\sim$45% in the BTH area, but then they say that low volatility products increased with a VOCs oxidation capacity factor of $\sim$1.3. But if we do a simple estimation, 1.3*0.55=0.71, which would mean the oxidation products would not increase. I'm not questioning the conclusion that oxidation products could increased, but there's just far too less data in this study to support this.

Response: The authors are appreciated for the reviewer's kind advice, and totally agreed that the direct evidence for the enhanced particle growth due to VOCs was not enough. We supplemented five kinds of VOCs (isoprene, benzene, toluene, C8 and C9 aromatics) derived from PTR-MS during the measurement (Fig. 1), which are the indicators of anthropogenic VOC and also could be oxidized to be HOMs, contributing to the growth process. The result showed C8 and C9 aromatics decreased by approximately 20% and 8% during LCD as compared with Pre_LCD, however, isoprene and toluene were slightly changed, benzene increased by approximately 21% during LCD period. It also suggested the VOCs we focused didn't show the reduction rate as 45% as Huang et al. (2020) reported in BTH region. Moreover, the major oxidants of VOCs (O3, OH and NO3) all increased during LCD period, indicating the possibility of enhanced HOMs participating the particle growth. Unfortunately, we cannot measure the oxidation products directly, e.g. by CIMS, in this study. And the simulation of oxidation product of VOCs by model is beyond the scope in this work. In the manuscript, we supplemented the measured VOCs data and detailed discussion, the oxidation products by proxy method was canceled as the reviewer suggested. But the oxidants of VOCs were discussed and we put more emphasis on the relationship between proxy sulfuric acid and NPF events.

2. Furthermore, it seems to me at least that authors does not understand the concept of

ELVOCs or HOMs correctly. It should be noted that the concept of "ELVOCs" should be used when we don't have information of the volatility, but only use "HOMS" when we are discussing oxidized VOCs. However, using k1*[OH]*[VOC]+k2*[OH]*[VOC] is certainly not acceptable as a proxy of HOMs. First, this represents the first generation product of oxidized VOCs, and even for a-pinene oxidation which is the most studied HOMs production pathway, this only produces very volatile OVOCs, and they won't contribute substantially for the growth of Aitken mode particles. Secondly, while this proxy was developed for an a-pinene rich boreal forest, the main VOCs in Beijing are aromatic, alkenes, and even the main BVOC are not a monoterpene but isoprene. So there needs to be far more discussion to settle down which is the main OVOC contributing as low volatile products, and multi-generation products instead of first-order products should be considered. Last but not least, in the wintertime in Beijing, the night is very long, and oxidation by NO3 should not be ignored.

Response: Thanks for the constructive comments. The authors have checked the manuscript to correct the terminology. The highly oxygenated organic molecules (HOMs) have been proved to be important for NPF. However, the direct measurement of HOMs is lacking in this work, and the simulation or determination of which VOCs can be oxidized to form low volatile products and contribute to the particle growth is complex and beyond the scope of this study. The effect of HOMs on nucleation and its following growth with be conducted further by applying CIMS in the further study. As explained above, the anthropogenic VOCs measurement data was supplemented in the text. The major pathways of HOMs formation are the oxidation by O3, OH and NO3 radicals (Atkinson and Arey, 2003). O3 increased by 80% during LCD period. We used Glob_R as a simple proxy of OH, and Glob_R increased by ∼24% during LCD as compared with pre-LCD. Thanks for the reviewer providing the new sight that the oxidation by NO3 is a key process of night chemistry. It has been reported the monoterpene oxidation initiated by NO3 played an important role for HOMs formation in boreal forest, especially in winter time (Yan et al., 2016; Kontkanen et al., 2016). In urban Beijing, NO3 oxidation of nocturnal BVOCs is also an important pathway of SOA formation

in summer time (Wang et al., 2018). However, the estimation of the multi-generation of VOCs product by NOx oxidation needs to conducted by applying model and more measurement data, which is not available in this work and the simple proxy can introduce large uncertainties. NO3 is predominantly formed by the reaction of NO2 with O3 (NO2+O3→NO3+O2 ), and we applied [NO2]*[O3] to estimate the NO3 production term. It showed [NO2]*[O3] term increased by ∼40% during LCD period, indicating the possibly enhanced oxidizing products of VOCs by NO3 during the nighttime. We also supplemented the discussion in the manuscript.

References:

Atkinson, R. and Arey, J.: Gas-phase tropospheric chemistry of biogenic volatile organic compounds: a review, Atmos. Environ., 37(Supplement No. 2): S197–S219, 2003.

Yan, C., Nie, W., Äijälä, M., Rissanen, M. P., et al.: Source characterization of highly oxidized multifunctional compounds in a boreal forest environment using positive matrix factorization, Atmospheric Chemistry and Physics, 16(19): 12715-12731, 2016.

Kontkanen, J., Paasonen, P., Aalto, J., Bäck, J., Rantala, P., Petäjä, T. and Kulmala, M.: Simple proxies for estimating the concentrations of monoterpenes and their oxidation products at a boreal forest site, Atmos. Chem. Phys., 16(20): 13291-13307, 2016.

Wang, H., Lu, K., Guo, S., Wu, Z., Shang, D., Tan, Z., Wang, Y., Le Breton, M., Lou, S., Tang, M., Wu, Y., Zhu, W., Zheng, J., Zeng, L., Hallquist, M., Hu, M. and Zhang, Y.: Efficient N2O5 uptake and NO3 oxidation in the outflow of urban Beijing, Atmospheric Chemistry and Physics, 18(13): 9705-9721, 2018.

3. The authors claim that nucleated particles can grow to CCN size and contribute to particle mass during haze events. But overall, there is very less discussion of investigation of the particle growing form nucleation mode to accumulation mode, which means that quantitative understanding is lacking. For Fig7., if we take 6th Feb for example, it seems that the growth from aitken to accumulation mode comes from growth of pre-existing particles. And even though the growth of particle number concentration seems to terminate by 7th Feb, it seems that the PM2.5 mass still increases rapidly. At noon 8th Feb, it looks like there is a new polluted air mass coming, leading for stronger pollution.

Response: The 48 hours back-trajectory analysis from February, 4th-14th, corresponding to the polluted case study period (Fig. 2), which is also supplemented in the manuscript. The back trajectories originated from northwest from February 4th to 10th, corresponding to the dry and clean air masses. However, from February 11th to 13th, the southwesterly air masses were dominated and favored the accumulating of the particles, resulting in the high concentration level of PM. The NPF events both occurred on February 4th and 5th, producing high number concentration of nucleated particles. On 5th and 6th, the air masses passed through Tianjin, which was a megacity in the southeast of Beijing, containing the anthropogenic gases and could favor the NPF growth process. The nucleated particles grew into the larger sizes in the following days until February 10th. On Feb 7th to 9th, the air masses all originated from northwest, and the variation of PM2.5 could be caused by the planetary boundary layer (PBL) mixing on Feb 8th. Moreover, the variation of local wind could also disturb the growth process. The PM2.5 normalized by CO also showed an increasing trend from February 4th to 10th, indicating a strong secondary aerosol formation.

4. I think the paper can be resubmitted by putting more effort on nucleation and early growth by sulfuric acid. The NAIS measurement seems to work good, and could be discussed more in depth. To explain the growth driven by oxidized VOCs, either support by measurement of chemical composition or a chemical mechanism model is needed.

Response: (1) The influence of sulfuric acid on the growth process was further analyzed as the reviewer suggested. Based on the NAIS data of neutral particle mode, the hourly mean geometric mean diameter of nucleation mode (Dp,nuc) was fitted to show the growth process. The result showed much higher proxy sulfuric acid concentration [H2SO4] during the LCD and post-LCD period, as compared with the pre-LCD period (Fig. 3). It also revealed that in the initial growth process (Dp,nuc< 5 nm), Dp,nuc increased positively with [H2SO4]. Furthermore, GR in the size range of 3-5 nm was slightly higher during LCD and post-LCD (0.72 nm/h), as compared with pre-LCD (0.60 nm/h), indicating the enhanced effect of sulfuric acid on the initial growth of the nucleated particles. However, when the nucleated particles grew into the larger sizes (> 5 nm), [H2SO4] decreased probably related with the weaken solar radiation wintertime, which could not explain the continuous growth and the VOCs could be the main contributor. As the reviewer recommended, we should have more discussion about NAIS data in details. The mean time evolution of Dp, nuc of neutral particles (Dp,nuc,par) and charged ions (Dp,nuc,ion) during the NPF events was given in Fig. 4. It showed Dp,nuc,ion grow faster than Dp,nuc,par, especially for the sizes below 10 nm, depending on the growth rate in each time interval ((Dp,nuc,t1- Dp,nuc,t2)/Δt, Δt = 1 h). The enhanced growth rate factor (GRp,nuc,ion/ GRp,nuc,par) ranged from 1.1 to 1.7, with the average of 1.38+-0.34 during the entire particle growth process and higher (∼2.0) for the initial size of 2–5 nm. (2) The times series of isoprene and major C6–C9 VOCs observed in this study by PTR-MS and the PDF distribution were given in Fig. 1. These VOC gases are too volatile to participate in nucleation or growth, they are good indicators of anthropogenic VOC plumes (Dai et al., 2017). It is possible that these plumes contained high concentrations of ammonia, amines, or HOMs produced from these VOCs, which are potential drivers of strong local NPF events. As compared with Pre_LCD period, the mean average of isoprene and toluene were slightly changed during LCD, C8 and C9 decreased by 20% and 8%, respectively, and the mixing ratio of benzene increased by approximately 21% during LCD period. However, the oxidized VOCs (HOMs) are difficult to be evaluated in this work.

Dai, L., Wang, H., Zhou, L., An, J., Tang, L., Lu, C., Yan, W., Liu, R., Kong, S., Chen, M., Lee, S. and Yu, H.: Regional and local new particle formation events observed in the Yangtze River Delta region, China, J. Geophys. Res., 122(4): 2389-2402, DOI: 10.1002/2016jd026030, 2017.

Minor comments:

1. The fitting coefficients for H2SO4 proxy should not used the same as the measurement in Finland. For Beijing, there's a paper by Lu et al (2019). Note that the effect is nonlinear and will effect trend in Fig 4-5. And if Global Radiation is used instead of UVB for H2SO4, it should be stated as it was done for OH. UVB is a fraction of Global Radiation, therefore a new coefficient should be used. Lu, Y., Yan, C., Fu, Y., Chen, Y., Liu, Y., Yang, G., Wang, Y., Bianchi, F., Chu, B., Zhou, Y., Yin, R., Baalbaki, R., Garmash, O., Deng, C., Wang, W., Liu, Y., Petäjä, T., Kerminen, V.-M., Jiang, J., Kulmala, M., and Wang, L.: A proxy for atmospheric daytime gaseous sulfuric acid concentration in urban Beijing, Atmos. Chem. Phys., 19, 1971–1983, https://doi.org/10.5194/acp-19-1971-2019, 2019.

Response: Thanks for the constructive comments, we re-calculated proxy H2SO4 by Lu et al., (2019) and did the reanalysis. In the new calculation of [H2SO4] in Beijing, we chose proxy equation number 2 (Eq. 1) and 7 (Eq. 2) as recommended by Lu et al. (2019), to represent the simplest and most accurate method, respectively.

$[H2SO4] = 280.05 \times UVB^{0.14} \times [SO2]^{0.40}$ (1)

$[H2SO4] = 0.0013 \times UVB^{0.13} \times [SO2]^{0.40} \times CS^{(-0.17)} \times ([O3]^{0.44} + [NOx]^{0.41})$ (2)

And the UVB was derived by 0.008% * Glob_R, based on the previous study that the monthly average of the ratio of UVB to global radiation (Glob_R) ranged from 0.007 to 0.017% in Beijing (Hu et al., 2013). The average ratio of January and February (0.008%) was applied. However, the results derive by N2 and N7 method (Lu et al., 2019) showed a clear difference, indicating the large uncertainty of the proxy method. And for several NPF events, the elevated concentration of sulfuric acid was not observed by N2 and N7 method. The main reason could be the role of CS was underestimated. In the previous study in Beijing (Lu et al., 2019), the covariance of CS and SO2 was found (correlation coefficient R=0.83) that offset the dependence of sulfuric acid on CS. However, during the measurement in our study, a special period of emission sharply decreased, R is 0.45 between SO2 and CS. As a compromise, we also referred the proxy method develop in boreal forest (Petäjä et al., 2009). The average value of three proxy method was applied to analyze the variation of sulfuric acid and its relationship with NPF events (Fig. 5).

2. I didn't found the OH proxy used here in Petäjä et al (2009). Please make sure the right reference is cited.

Response: the correct citation should be "Nieminen, T., Keronen, P., Asmi, A., Petäjä, T., maso, M. D., Kulmala, M. and Kerminen, V.-M.: Trends in atmospheric new-particle formation: 16 years of observations in a boreal-forest environment, Boreal Envrion. Res., 19 (suppl. B): 191–214, 2014."

3. There are spelling mistakes and grammar errors, try to find an english expert to fix them all, eg: The number concentration of Aitken mode particles (∼25-100nm) should also decreased as expected-> The number concentration of Aitken mode particles (∼25-100nm) decreased as expected or The number concentration of Aitken mode particles (∼25-100nm) decreased as they should.

Response: The spelling and grammar have checked through the manuscript, which has been language edited by the English native speakers. The sentence that reviewer mentioned has been revised to be "The number concentration of Aitken mode particles (∼25-100 nm), which is related with the traffic emission (Deventer et al., 2018) is also expected to decrease."

4. Use the term oxidizing capacity consistently, replace all "oxidization capacity"

Response: It has been revised as the reviewer suggested.
* * *
[Figure]

**Fig. 1.** Time series of isoprene, benzene, toluene, C8 and C9 aromatics (a-e) during January 5 to February 15, and the probability distribution function of mixing ration of each VOC component (f-j), respective

[Figure]

**Fig. 2.** The 48 h back-trajectory arriving at CAMS at 12:00 local time from February 4 to 14, the star indicating the measurement site (CAMS) in Beijing.

[Figure]

**Fig. 3.** Scatter plot between geometric mean diameter of nucleation mode (Dp,nuc) and the proxy sulfuric acid. The grey dots and crosses represent the NPF events during Pre_LCD, LCD/Post_LCD, respectively. The

[Figure]

**Fig. 4.** The time evolution of geometric mean diameter of nucleation mode (Dp,nuc) of neutral particle and positive charged ions during the NPF events. The circle and bar present the mean value and the standar

[Figure]

**Fig. 5.** The sulfuric acid concentrations derived by different proxy equations. The red and orange lines indicate the result by N2 and N7 method by Lu et al., 2019, and blue line indicates the method recommend

---

## Author Response (AR1)

Response to RC1

In this study, the authors compare observations before, during and after the lockdown period in China during January and February 2020. They observe an enhancement of the nucleation and growth process of nanoparticles during the lockdown in Beijing. From this, they conclude that these findings were caused by the lockdown period, mainly due to lower concentration of Aitken mode aerosols that reduces the condensation sink. In contrast, accumulation mode particles increased and caused pollution events, due to new particle formation events with subsequent growth. This enhanced particle nucleation and growth is attributed to enhanced values of $H_2SO_4$ and VOC oxidation products, which were calculated from available data.

The subject of this study is certainly suited for ACP.

However, I have two major concerns regarding this study that need to be addressed in a revised version. These concerns are a) meteorological representativeness and b) statistical significance. I explain my concerns in detail in the following.

**Response**: The authors thank the reviewer's comments and try our best to address the issues point-by point.

Major issues:

a) Meteorological representativeness

A major problem when comparing air pollution data from different periods is the influence of meteorology. This needs to be considered to ensure that the observed differences are not coincidence. High and low pressure systems may prevail for a certain time, leading to differences in cloud coverage and thereby to enhanced or reduced radiation. Wind speed and direction influences transport of pollution from sources, either regional or even from long distances. Meteorological parameters are available. I suggest conducting something like a 5-year climatology of the available parameters to check the variability of the atmosphere and the representativeness of the Jan-Feb 2020 period, especially the lockdown period, compared to the same period in previous years. In line 40-41, the authors state that "Furthermore, particle accumulation was favored by stagnant airflow and vertical meteorological conditions during LCD (Zhong et al., 2020)." So apparently, they are aware of unusual meteorological conditions during the LCD period. But, since the Zhong et al. reference is still in preparation, the reader can't retrieve this additional but important information.

**Response**: Thanks for the reviewer's constructive comment. (1) The meteorological parameters during LCD period, January and February in 2020, as well as the average conditions of January and February in 2016-2020 were analyzed and the diurnal patten was given (Fig. S1). It showed much higher RH, lower wind speed, slightly higher temperature and lower pressure during LCD and January and February 2020, than that of 5-year climatology average condition (January and February in 2016-2020). The anomaly of monthly mean sea level pressure in January and February between 2020 and 2016-2020 was analyzed based on the ECMWF reanalysis dataset (ERA5, https://cds.climate.copernicus.eu/), as given in Fig. S2. It showed negative anomaly in Beijing-Tianjin-Hebei Province, indicating the air pressure decreased in

January and February in 2020, as compared with the corresponding period of the 5-year climatology. The local air convergence resulted in high RH and low wind speed, which favored for the air pollutants accumulating. (2) In line 40-41, we removed the Zhong et al., 2020 in the manuscript and referred the previous publication (Zhong et al., 2018) also conducted in Beijing to address the meteorological effect on the air pollution formation. The discussion was supplemented in the section of "3.1 The meteorological conditions".

[Figure]

Fig. S1. The mean diurnal pattern of meteorological parameters, including temperature (a), RH (b), sea level pressure (c) and wind speed (d) during LCD period (January 24-February 16, 2020), January and February in 2020 and in 2016-2020. The solid circles and bars represent the mean value and the standard deviation, respectively.

[Figure]

Fig. S2. The anomaly of monthly mean sea level pressure in January and February between 2020 and 2016-2020. The data are from the ERA5 ECMWF reanalysis dataset (https://cds.climate.copernicus.eu/).

b) Statistic significance
The changes in NPF event frequency seem not to be significant, because the number of NPF events is small. The authors report on differences of "10 out of 23 days (43%)", "8 out of 24 days (33%)", and "5 out of 13 days (38%)". These are small numbers. A

simple estimation based on Poisson statistics suggests that these differences may not be significant. Here a detailed statistical analysis has to be presented, and it may be that the results will be that these differences are not significant.

Similar analyses have to be done for pollutants NO2 and SO2, because Fig. 3 shows that both are highly variable during the pre-LCD, LCD, and post-LCD periods. The linear regressions between H2SO4, J2, and GR are significant (Fig 5), but that's not new. And since H2SO4 is mainly calculated from global radiation, the meteorological influence on this parameter is high. The different growth rates when comparing pre-LCD, LCD and post-LCD for the different size ranges presented in Fig 6 may also not be significant, regarding the error bars.

**Response:** (1) The Poisson statistics was conducted for NPF event occurrence probability for pre-LCD, LCD and post-LCD period, respectively, as given in Fig. S3. It showed almost the same NPF event occurrence probability as compared with pre-LCD and LCD period, but fewer NPF event during Post_LCD. (2) The probability density function (PDF) was given for $SO_2$, $NO_2$ and $O_3$ during Pre_LCD, LCD and Post_LCD, respectively, as given in Fig. S4. It showed significant decreasing trend of $NO_2$, whereas increasing trend of $O_3$ as compared with Pre_LCD and LCD/Post_LCD. However, the variation of $SO_2$ among different periods was not clear, as the $SO_2$ concentration remained low due to the emission control these years. The PDF of gases pollutants and the detailed discussion have been added in the text and the figure is given in the supplementary materials. (3) $H_2SO_4$ proxy was derived by three methods and the average value was applied for discussion, in order to minimize the uncertainties. (4) the growth rate (*GR*) was discussed for different size range 2-5 nm, 5-10 nm and >10 nm, respectively. It showed $GR_{2-5nm}$ and $GR_{>10nm}$ were generally higher during LCD/Post_LCD as compared with Pre_LCD, however, the difference of $GR_{5-10\ nm}$ is not clear.

[Figure]

Fig. S3 Poisson distribution of NPF event occurrence frequency during pre-LCD, LCD and post-LCD, respectively.

[Figure]

[Figure]

[Figure]

Fig. S4. The probability density function (PDF) of SO₂, NO₂ and O₃ concentration during pre-LCD, LCD and post-LCD, respectively.

Minor comments

Line 20: Please explain the meaning of J2 also in the Abstract.

**Response**: It has been revised to be "higher formation rate of 2 nm particles ($J_2$) and the subsequent growth rate (GR)".

Line 34-35: Reformulate this sentence to: The number of Aitken mode particles (d~ 25 – 100 nm), which is related to traffic emissions (Deventer et al., 2018) is also expected to decrease.

**Response**: It has been revised in the text.

Line 38: Change "Air pollution is driven by the enhancement of secondary particles, " to "Secondary particles contribute significantly to air pollution"

**Response**: It has been revised in the text.

Line 41: The reference to Zhong et al. in preparation is not sufficient. If you do not want to show these data now, then I suggest to wait until the Zhong et al. paper is submitted as well. The meteorological situation (e.g. the inversion layer) is of great importance (see my major comment above).

**Response**: we removed the Zhong et al., 2020 in the manuscript and referred the previous publication (Zhong et al., 2018) also conducted in Beijing to address the meteorological effect on the air pollution formation. The discussion was supplemented in the section of "3.1 The meteorological conditions".

Zhong, J., Zhang, X., Dong, Y., Wang, Y., Liu, C., Wang, J., Zhang, Y. and Che, H.: Feedback effects of boundary-layer meteorological factors on cumulative explosive growth of PM2.5 during winter heavy pollution episodes in Beijing from 2013 to 2016, Atmospheric Chemistry and Physics, 18(1): 247-258, DOI: 10.5194/acp-18-247-2018, 2018.

Line 107 – 113: H2SO4 estimation: This is a very rough estimation. What are usual values for k, and what is the dimension of k? In Figure 4 no units are given on the right y-axis, but shouldn't that be cm-3?

**Response**: The concentration of $H_2SO_4$ was not measured directly in this study and different proxy methods were referred to derive the proxy sulfuric acid. A method (Eq. 3) depends on the global radiation (Glob_R), $SO_2$ and condensation sink ($CS$), and is developed according to the previous study conducted in a forest site, Hyytiälä, Finland (Petäjä et al., 2009).

$$[H_2SO_4] = \frac{k \times Glob\_R \times SO_2}{CS} \tag{1}$$

where $k$ is empirically derived factor and well correlated with Glob_R (k=1.4×10$^{-7}$×Glob_R$^{-0.7}$, unit: m$^2$ W$^{-1}$ s$^{-1}$). The proxy equation is site-specific due to the different atmospheric conditions. In the polluted atmosphere, such as in Beijing, several proxy methods were also constructed based on a number of available atmospheric parameters (Lu et al., 2019). In this study, the simplest proxy (Eq. 4) and best performance proxy (Eq. 5) are adopted to derive the proxy sulfuric acid.

$$[H_2SO_4] = 280.05 \times UVB^{0.14} \times [SO_2]^{0.40} \tag{2}$$

$$[H_2SO_4] = 0.0013 \times UVB^{0.13} \times [SO_2]^{0.40} \times CS^{-0.17} \times ([O_3]^{0.44} + [NO_x]^{0.41})$$
(3)

$[H_2SO_4]$ is the gaseous sulfuric acid with the unit of molecule cm$^{-3}$. $[SO_2]$, $[O_3]$ and $[NO_x]$ is the concentration of sulfur dioxide, ozone, and nitrogen oxides, with the unit of molecule cm$^{-3}$. UVB is the intensity of ultraviolet radiation $b$ in W m$^{-2}$. $CS$ is the condensation sink, which describes how fast the vapor molecules condense on the existing particles (Dal Maso et al., 2002), with the unit of s$^{-1}$. The proxy method has been validated by comparing the measured sulfuric acid with a high correlation coefficient of 0.86 (Lu et al., 2019), based on the field campaign conducted approximately 2 km away from CAMS site. In this work, the direct measurement of UVB was not available. However, it had been reported by Hu et al. (2013) that the monthly average of the ratio of UVB to global radiation (Glob_R) ranged from 0.007 to 0.017% in Beijing. And in this study, the average ratio of January and February (0.008%) was applied to derive UVB by 0.008%× Glob_R.

Lines 123-130: So, to infer VOC oxidation capacity, you don't have OH measurements, thus you approximate OH, but you don't have UVB, so you approximate by global radiation. This seems like many uncertainties. Can you comment a little more on the uncertainties and the influence they might have on your interpretation?

**Response**: As discussed above, the UVB was derived by 0.008% × Glob_R, based on the previous study that the monthly average of the ratio of UVB to global radiation (Glob_R) ranged from 0.007 to 0.017% in Beijing (Hu et al., 2013). The average ratio of January and Feburary (0.008%) was applied in this study for calculating $[H_2SO_4]$.

Line 126, Equ. 6: There is a ratio of two numbers (8.4e-7/8.6e-10). Are these numbers rate coefficients that should have units? Or are they just empirical fit parameters? If they are just parameters, you can replace them by 9.8e2.

**Response**: The equation has been removed from the manuscript.

Line 132-135: What are the exact criteria to define the NPF events?

**Response**: It has been revised to "NPF events are identified and different nucleation types are characterized based on the daily evolution of particle number size distribution (PNSD). The burst of nucleation mode particles with diameter ≤ 25 nm appeared in the PNSD, and the burst should prevail over a few hours with clear growth process (Dal Maso et al., 2005)."

Line 140-142 & 178: As already commented above: "The NPF event occurred on 10 out of 23 days (43%) during pre-LCD, 8 out of 24 days (33%) in LCD, and 5 out of 13 days (38%) in post-LCD, respectively". The frequencies of 43% or 33% are based on a very low number of events. Please add the total number of events to the table, and calculate Binomial or Poisson statistics for these numbers. It may be that the differences are too small to be significant, just by the small number of events.
**Response**: We gave the number of NPF days and total available measurement days in table 1, instead of the NPF frequency. The Poisson statistics was conducted for NPF event occurrence probability for pre-LCD, LCD and post-LCD period, respectively, as given in Fig. S1. It showed fewer NPF events with higher probability as compared with pre-LCD and LCD period.

Line 150: Please refer to Fig 2a here and replace "…were discussed in detail." by "are discussed in detail in the following". Otherwise it is hard for the reader to follow this discussion.
**Response**: The sentence has been revised to "were given in Fig. 2a and discussed in detail in the following."

Line 170/171: Refer to Fig 2b here.
**Response**: It has been revised in the text.

Line 174: Refer to Fig 2c.
**Response**: It has been revised in the text.

Line 180: What is J3? Should it read J2? Or does this refer to measurements from 2015 where 3 nm was the lower size? If so, please explain.
**Response**: A sentence was given to explain the meaning of $J_3$ in Line 185 "$J_3$ referred the formation rate at 3 nm calculated from the particle number concentration of 3–4-nm particles by Eq. (1), as the lowest detection limit of SMPS applied in 2015 and 2010-2013 campaign was 3 nm."

Line 182-184: "in this study", "The previous study". Please make clear which study is which.
**Response**: "in this study" refers to the results of this manuscript and "the previous study" are the references we cited (Lehtipalo et al., 2018, Yan et al., 2020). The sentences have been revised to be "The daily mean value of $NO_2$ decreased by ~35% and $SO_2$ decreased by ~13%, whereas $O_3$ increased by 80% during LCD as compared to pre-LCD in this work (Fig. 3). Previous studies had indicated that $NO_x$ suppressed NPF events by

influencing the formation of highly oxygenated organic molecules (HOMs), which participated in nucleation and initial particle growth (Lehtipalo et al., 2018; Yan et al., 2016; 2020)."

Line 191: " As discussed separately for LCD and pre-LCD during the NPF event occurrence (9:00–16:00 LT),…". Which NPF event are you talking about? You specify the time period 9:00 – 16:00 LT. But Fig. 4 shows a time series of the whole campaign. Individual events can not be seen here. The whole paragraph line 191 until line 200 can not be understood, because you refer to one NPF event that is not shown. Which day is that? I assume that this text describes an event that was discussed in Huang et al 2020, but if that is so, this discussion does not belong here in this paper. Please include a time series of the measured parameters for this event here or skip this paragraph.
**Response**: "the NPF event occurrence (9:00–16:00 LT)" indicates all the NPF events usually occur during the daytime (9:00-16:00 LT), not refers to a specific NPF event. And the paragraph line 191 until line 200 are the discussion based on Fig. 4, describing the general characteristics of all NPF events during the measurement. We reorganized this paragraph to make it clearer.

Line 201: " : : :was indicated by different VOC_ox,capacity levels" seems not to be the right expression here. I suggest to rephrase the whole sentence: "Both the H2SO4 proxy and the VOX_ox,capacity were correlated to J_2,tot and to GR (Fig 5)."
**Response**: The sentence has been revised according to reviewer's comment.

Line 205-207: This is again a result from another study. Make that clear at the beginning of the sentence, like "Stolzenburg et al. (2020) showed that sulfuric acid could not explain: : :."
**Response**: the sentence has been revised to "Stolzenburg et al. (2020) revealed that sulfuric acid played an important role in smaller growth processes from 2–10 nm, however, could not explain condensational growth when the nucleated particles overcame 10 nm"

Lines 208-210: If H2SO4 contributed more to the nucleation process and organic vapours to the growth, would you not expect to see a better correlation between H2SO4 and J2 than between H2SO4 and GR, and similarly a better correlation between VOC_ox,cap and GR than between VOC_ox,cap and J2?
**Response**: The proxy sulfuric acid was re-calculated based on the reviewer's comment, and the effect of sulfuric acid on formation rate ($J_2$) and initial growth rate ($GR$). The influence of $H_2SO_4$ on $J_2$ and $GR$ was re-evaluated, and it showed a slightly higher correlation coefficient ($R$) of $J_2$ (R=0.62) than the $GR$ (R=0.45). However, the oxidation product could not be estimated simply, but also discussed in the manuscript. The estimation of VOCs oxidizing capacity was removed in the manuscript, as the proxy method was not reasonable as the reviewer suggested. However, the direct measurement data of 5 major kinds of VOCs is supplemented and the variation is discussed. The major oxidants of VOCs were also found to be elevated during LCD,

indicating the possibility of enhanced oxidation products of VOCs that promoted the nucleation and growth process.

Lines 214-215: For the size range 5-10 nm, there is no significant difference between ions and neutral particles. Especially the yellow bars for 5-10 nm (Fig. 6) have almost exactly the same height. What are the error bars and what is their meaning? This should explained in the caption of Fig 6.

**Response**: The detailed growth process of the nucleated particles and ions on NPF days was given in Fig. S5. It showed $D_{p,nuc,ion}$ grow faster than $D_{p,nuc,par}$, especially for the sizes below 5 nm, depending on the growth rate in each time interval (($D_{p,nuc,t1}$-$D_{p,nuc,t2}$)/$\Delta t$, $\Delta t$ = 1 h). The enhanced growth rate factor ($GR_{p,nuc,ion}$/ $GR_{p,nuc,par}$) ranged from 1.1 to 2.0, with the average of 1.38±0.34. The enhancement was higher in sub-10 nm particles, whereas it decreased as the particles grew to larger sizes. In addition, the histogram and error bars represent the mean value and standard deviation, respectively, which has been clarified in Fig. 6 caption.

[Figure]

Fig. S5. The time evolution of geometric mean diameter of nucleation mode ($D_{p,nuc}$) of neutral particle and positive charged ions during the NPF events. The circle and bar present the mean value and the standard deviation.

Line 220: What is the enhancement factor? How is it calculated?

**Response**: The enhancement effect (*EF*) describes the dipole-charge interaction on the growth of charged clusters (Nadykto and Yu, 2003). The charge effect of ions on the NPF include accelerated rates of vapor condensation and particle coagulation, as well as the charge recombination (Yu and Turco, 1998; 2000). *EF* for the pure species participating NPF event (eg, sulfuric acid, VOCs) can be calculated as the below equation (Nadykto and Yu, 2003):

$$EF = 1 + \frac{2lE(r_p + r_m)L(\frac{lE(r_p + r_m)}{kT} + \alpha\varepsilon_0 E^2(r_p + r_m))}{3kT} \quad (4)$$

*EF* depends on temperature (*T*), size of charged particles ($r_p$), and microphysical properties (dipole moment: *l*, polarizability: α, and size: $r_m$) of vapor molecules. For

sulfuric acid molecules, *EF* could be 10 for ions with ~0.5 nm but decreased quickly to 2 for uptake by a charged particle of ~2 nm, at T=300 K (Nadykto and Yu, 2003). In this work, we used enhancement factor ($GR_{ion}/GR_{par}$) to denote the net influence of charge on the particle growth. In order to differentiate with EF defined by equation 4, we used "enhanced growth rate factor" to denote $GR_{ion}/GR_{par}$. We also re-organized the section "effect of charge ions" to make it easier to understand.

References:

Nadykto, A. B. and Yu, F.: Uptake of neutral polar vapor molecules by charged clusters/particles: Enhancement due to dipole-charge interaction, J. Geophys. Res., 108(D23), DOI: 10.1029/2003jd003664, 2003.

Yu, F. and Turco, R. P.: The formation and evolution of aerosols in stratospheric aircraft plumes: Numerical simulations and comparisons with observations, Journal of Geophysical Research: Atmospheres, 103(D20): 25915-25934, DOI: 10.1029/98jd02453, 1998.

Yu, F. Q. and Turco, R. P.: Ultrafine aerosol formation via ion-mediated nucleation, Geophys. Res. Lett., 27(6): 883-886, DOI: 10.1029/1999gl011151, 2000.

Line 223: Replace "effect of the charger" by "effect of charge"
**Response:** It has been revised in the text.

Lines 228-231: It would be helpful to add PM2.5 to one of the time series in Fig. 3, and to add the numbers of the NPF events in Fig 3 and 4 instead of or additional to the crosses. When I count the NPF events marked by the crosses, I find that event #9 is on January 23.
**Response**: The figures has been revised.

[Figure]

Fig. S6. Concentration level of PM$_{2.5}$ mass concentration (a), and precursors (b), including NO$_2$, SO$_2$, and O$_3$ during the measurement period. The circle and bar indicate the mean and standard deviation, respectively; NPF days are marked with continuous numbers 1-25.

Lines 233-238: Please include a graph showing PM2.5/CO. Please also state clearly how pollution periods were identified.

**Response**: The discussion from lines 233-238 are referred to Fig. 7, which have contained $PM_{2.5}$/CO in subplot (b). We revised the sentence as "Two principal pollution episode formation stages were identified according to variations in the $PM_{2.5}$ mass concentration dividing by CO ($PM_{2.5}$/CO), as indicated in Fig. 7b.". The pollution episodes were defined as the daily mean value of $PM_{2.5}$ mass concentration exceeding 75 μg m$^{-3}$, which is the criterion value of the second grade of air quality in China. We have added this sentence in the text.

Line 238-240: This sentence doesn't make sense. Maybe the "and" before "unfavorable" needs to be deleted?

**Response:** It has been revised in the text, "and" has been deleted.

Whole Section 3.4: What is the message of the section 3.4? The reader can not see the pollution events if there is no graph showing PM2.5, or CO, or both. Are there any conclusion drawn by section 3.4? It all seems very speculative. Meteorological conditions are mentioned as one possible reason for this pollution event, but it is not investigated by trajectory and emission source locations.

**Response:** The discussion of section 3.4 is referred to Fig. 10, including the evolution of PNSD, $PM_{2.5}$ and its normalization by CO, which has been clarified in the text. The meteorological factor including wind direction, speed and relative humidity has been given in Fig. 10. Furthermore, the back-trajectory analysis from Feb, 4$^{th}$-14$^{th}$, corresponding to the study period in Fig. 10, is also supplemented (given as Fig. 11 in the manuscript). The back trajectories originated from northwest from February 4$^{th}$ to 10$^{th}$, corresponding to the dry and clean air masses. However, from February 11$^{th}$ to 13$^{th}$, the southwesterly air masses were dominated and favored the accumulating of the particles, resulting in the high concentration level of particle matter.

A paragraph of "2.5 Back trajectory analysis" is given in the section of "2. Method"

In order to reveal the meteorological condition during the pollution case formation, the 48 h backward trajectories arriving at CAMS stie were calculated at 12:00 Local time, terminating at the height of 500 m above ground level by applying the Trajstat Software, combined with HYSPLIT 4 model (Hybrid Single-Particle Lagrangian Integrated Trajectory) and using the NCEP GDAS (Global Data Assimilation System) data with 1°×1° resolution (Draxler and Hess, 1998, Wang et al., 2009).

[Figure]

Fig. S7. The back-trajectory arriving at CAMS at 12:00 local time from Feb 4[th] to 14[th], the star indicating the measurement site (CAMS) in Beijing.

**Response to RC2**

This study presents observational results of particle and ions number concentration in Beijing, China during Jan. 24 - Feb. 14, 2020. This period represent the Chinese New year and the COVID-19 lockdown period, therefore suitable for understanding the influence of reduced emission to nanoparticle formation and growth. Although the data set is novel and the main research question it wants to discuss sounds interesting, there are too many important values and discussion in the text which are based on guessing. Overall, I don't recommend this paper to be considered to be published before major reconstruction.

**Response**: The authors thank the reviewer's comments and try our best to address the issues point-by point.

1. I would say that the current evidence (No measured or modelled VOCs) does not allow the major conclusion on VOC contribution of particle growth of particles of 10nm-100nm. The only mentioning of VOCs is that reduces by ~45% in the BTH area, but then they say that low volatility products increased with a VOCs oxidation capacity factor of ~1.3. But if we do a simple estimation, 1.3*0.55=0.71, which would mean the oxidation products would not increase. I'm not questioning the conclusion that oxidation products could increased, but there's just far too less data in this study to support this.

**Response**: The authors are appreciated for the reviewer's kind advice, and totally agreed that the direct evidence for the enhanced particle growth due to VOCs was not enough. We supplemented five kinds of VOCs (isoprene, benzene, toluene, C8 and C9 aromatics) derived from PTR-MS during the measurement, which are the indicators of anthropogenic VOC and also could be oxidized to be HOMs, contributing to the growth process. The result showed C8 and C9 aromatics decreased by approximately 20% and 8% during LCD as compared with Pre_LCD, however, isoprene and toluene were slightly changed, benzene increased by approximately 21% during LCD period. It also suggested the VOCs we focused didn't show the reduction rate as 45% as Huang et al. (2020) reported in BTH region. Moreover, the major oxidants of VOCs ($O_3$, OH and $NO_3$) all increased during LCD period, indicating the possibility of enhanced HOMs participating the particle growth. Unfortunately, we cannot measure the oxidation products directly, e.g. by CIMS, in this study. And the simulation of oxidation product of VOCs by model is beyond the scope in this work. In the manuscript, we supplemented the measured VOCs data and detailed discussion, the oxidation products by proxy method was canceled as the reviewer suggested. But the oxidants of VOCs were discussed and we put more emphasis on the relationship between proxy sulfuric acid and NPF events.

[Figure]

Fig. S1. Time series of isoprene, benzene, toluene, C8 and C9 aromatics (a-e) during January 5 to February 15, and the probability distribution function of mixing ration of each VOC component (f-j), respectively.

2. Furthermore, it seems to me at least that authors does not understand the concept of ELVOCs or HOMs correctly. It should be noted that the concept of "ELVOCs" should be used when we don't have information of the volatility, but only use "HOMS" when we are discussing oxidized VOCs. However, using k1*[OH]*[VOC]+k2*[OH]*[VOC] is certainly not acceptable as a proxy of HOMs. First, this represents the first generation product of oxidized VOCs, and even for a-pinene oxidation which is the most studied HOMs production pathway, this only produces very volatile OVOCs, and they won't contribute substantially for the growth of Aitken mode particles. Secondly, while this proxy was developed for an a-pinene rich boreal forest, the main VOCs in Beijing are aromatic, alkenes, and even the main BVOC are not a monoterpene but isoprene. So there needs to be far more discussion to settle down which is the main OVOC contributing as low volatile products, and multi-generation products instead of first-order products should be considered. Last but not least, in the wintertime in Beijing, the night is very long, and oxidation by NO3 should not be ignored.

**Response**: Thanks for the constructive comments. The authors have checked the manuscript to correct the terminology. The highly oxygenated organic molecules (HOMs) have been proved to be important for NPF. However, the direct measurement of HOMs is lacking in this work, and the simulation or determination of which VOCs can be oxidized to form low volatile products and contribute to the particle growth is complex and beyond the scope of this study. The effect of HOMs on nucleation and its following growth with be conducted further by applying CIMS in the further study. As explained above, the anthropogenic VOCs measurement data was supplemented in the text. The major pathways of HOMs formation are the oxidation by $O_3$, OH and $NO_3$

radicals (Atkinson and Arey, 2003). $O_3$ increased by 80% during LCD period. We used Glob_R as a simple proxy of OH, and Glob_R increased by ~24% during LCD as compared with pre-LCD.

Thanks for the reviewer providing the new sight that the oxidation by $NO_3$ is a key process of night chemistry. It has been reported the monoterpene oxidation initiated by $NO_3$ played an important role for HOMs formation in boreal forest, especially in winter time (Yan et al., 2016; Kontkanen et al., 2016). In urban Beijing, $NO_3$ oxidation of nocturnal BVOCs is also an important pathway of SOA formation in summer time (Wang et al., 2018). However, the estimation of the multi-generation of VOCs product by $NO_x$ oxidation needs to conducted by applying model and more measurement data, which is not available in this work and the simple proxy can introduce large uncertainties. $NO_3$ is predominantly formed by the reaction of $NO_2$ with $O_3$ ($NO_2 + O_3 \rightarrow NO_3 + O_2$), and we applied [$NO_2$]×[$O_3$] to estimate the $NO_3$ production term. It showed [$NO_2$]×[$O_3$] term increased by ~40% during LCD period, indicating the possibly enhanced oxidizing products of VOCs by $NO_3$ during the nighttime. We also supplemented the discussion in the manuscript.

References:

Atkinson, R. and Arey, J.: Gas-phase tropospheric chemistry of biogenic volatile organic compounds: a review, Atmos. Environ., 37(Supplement No. 2): S197–S219, 2003.

Yan, C., Nie, W., Äijälä, M., Rissanen, M. P., et al.: Source characterization of highly oxidized multifunctional compounds in a boreal forest environment using positive matrix factorization, Atmospheric Chemistry and Physics, 16(19): 12715-12731, 2016.

Kontkanen, J., Paasonen, P., Aalto, J., Bäck, J., Rantala, P., Petäjä, T. and Kulmala, M.: Simple proxies for estimating the concentrations of monoterpenes and their oxidation products at a boreal forest site, Atmos. Chem. Phys., 16(20): 13291-13307, 2016.

Wang, H., Lu, K., Guo, S., Wu, Z., Shang, D., Tan, Z., Wang, Y., Le Breton, M., Lou, S., Tang, M., Wu, Y., Zhu, W., Zheng, J., Zeng, L., Hallquist, M., Hu, M. and Zhang, Y.: Efficient $N_2O_5$ uptake and $NO_3$ oxidation in the outflow of urban Beijing, Atmospheric Chemistry and Physics, 18(13): 9705-9721, 2018.

3. The authors claim that nucleated particles can grow to CCN size and contribute to particle mass during haze events. But overall, there is very less discussion of investigation of the particle growing form nucleation mode to accumulation mode, which means that quantitative understanding is lacking. For Fig7., if we take 6th Feb for example, it seems that the growth from aitken to accumulation mode comes from growth of pre-existing particles. And even though the growth of particle number concentration seems to terminate by 7th Feb, it seems that the PM$_{2.5}$ mass still increases rapidly. At noon 8th Feb, it looks like there is a new polluted air mass coming, leading for stronger pollution.

**Response**: The 48 hours back-trajectory analysis from February, 4$^{th}$-14$^{th}$, corresponding to the polluted case study period, which is also supplemented in the manuscript. The

back trajectories originated from northwest from February 4[th] to 10[th], corresponding to the dry and clean air masses. However, from February 11[th] to 13[th], the southwesterly air masses were dominated and favored the accumulating of the particles, resulting in the high concentration level of PM. The NPF events both occurred on February 4[th] and 5[th], producing high number concentration of nucleated particles. On 5[th] and 6[th], the air masses passed through Tianjin, which was a megacity in the southeast of Beijing, containing the anthropogenic gases and could favor the NPF growth process. The nucleated particles grew into the larger sizes in the following days until February 10[th]. On Feb 7[th] to 9[th], the air masses all originated from northwest, and the variation of $PM_{2.5}$ could be caused by the planetary boundary layer (PBL) mixing on Feb 8[th]. Moreover, the variation of local wind could also disturb the growth process. The $PM_{2.5}$ normalized by CO also showed an increasing trend from February 4[th] to 10[th], indicating a strong secondary aerosol formation.

[Figure]

Fig. S2. The 48 h back-trajectory arriving at CAMS at 12:00 local time from February 4 to 14, the star indicating the measurement site (CAMS) in Beijing.

4. I think the paper can be resubmitted by putting more effort on nucleation and early growth by sulfuric acid. The NAIS measurement seems to work good, and could be discussed more in depth. To explain the growth driven by oxidized VOCs, either support by measurement of chemical composition or a chemical mechanism model is needed.

**Response**: (1) The influence of sulfuric acid on the growth process was further analyzed as the reviewer suggested. Based on the NAIS data of neutral particle mode, the hourly mean geometric mean diameter of nucleation mode ($D_{p,nuc}$) was fitted to show the growth process. The result showed much higher proxy sulfuric acid concentration [$H_2SO_4$] during the LCD and post-LCD period, as compared with the pre-LCD period (Fig. S3). It also revealed that in the initial growth process ($D_{p,nuc} < 5$ nm), $D_{p,nuc}$ increased positively with [$H_2SO_4$]. Furthermore, $GR$ in the size range of 3-5 nm was slightly higher during LCD and post-LCD (0.72 nm/h), as compared with pre-LCD (0.60 nm/h), indicating the enhanced effect of sulfuric acid on the initial growth of the nucleated particles. However, when the nucleated particles grew into the larger sizes (> 5 nm), [$H_2SO_4$] decreased probably related with the weaken solar radiation wintertime,

which could not explain the continuous growth and the VOCs could be the main contributor.

[Figure]

Fig. S3. Scatter plot between geometric mean diameter of nucleation mode ($D_{p,nuc}$) and the proxy sulfuric acid. The grey dots and crosses represent the NPF events during Pre_LCD, LCD/Post_LCD, respectively. The purple and blue lines represent the mean conditions during Pre_LCD, LCD/Post_LCD. The vertical and horizontal bars represents the standard deviations of sulfuric acid and $D_{p,nuc}$.

As the reviewer recommended, we should have more discussion about NAIS data in details. The mean time evolution of $D_{p, nuc}$ of neutral particles ($D_{p,nuc,par}$) and charged ions ($D_{p,nuc,ion}$) during the NPF events was given in Fig. S4. It showed $D_{p,nuc,ion}$ grow faster than $D_{p,nuc,par}$, especially for the sizes below 10 nm, depending on the growth rate in each time interval (($D_{p,nuc,t1}$- $D_{p,nuc,t2}$)/Δt, Δt = 1 h). The enhanced growth rate factor ($GR_{p,nuc,ion}$/ $GR_{p,nuc,par}$) ranged from 1.1 to 1.7, with the average of 1.38±0.34 during the entire particle growth process and higher (~2.0) for the initial size of 2–5 nm.

[Figure]

Fig.S4. The time evolution of geometric mean diameter of nucleation mode ($D_{p,nuc}$) of neutral particle and positive charged ions during the NPF events. The circle and bar present the mean value and the standard deviation.

(2) The times series of isoprene and major C6–C9 VOCs observed in this study by PTR-MS and the PDF distribution were given in Fig. S1. These VOC gases are too volatile

to participate in nucleation or growth, they are good indicators of anthropogenic VOC plumes (Dai et al., 2017). It is possible that these plumes contained high concentrations of ammonia, amines, or HOMs produced from these VOCs, which are potential drivers of strong local NPF events. As compared with Pre_LCD period, the mean average of isoprene and toluene were slightly changed during LCD, C8 and C9 decreased by 20% and 8%, respectively, and the mixing ratio of benzene increased by approximately 21% during LCD period. However, the oxidized VOCs (HOMs) are difficult to be evaluated in this work.

Dai, L., Wang, H., Zhou, L., An, J., Tang, L., Lu, C., Yan, W., Liu, R., Kong, S., Chen, M., Lee, S. and Yu, H.: Regional and local new particle formation events observed in the Yangtze River Delta region, China, J. Geophys. Res., 122(4): 2389-2402, DOI: 10.1002/2016jd026030, 2017.

**Minor comments:**

1.  The fitting coefficients for H2SO4 proxy should not used the same as the measurement in Finland. For Beijing, there's a paper by Lu et al (2019). Note that the effect is nonlinear and will effect trend in Fig 4-5. And if Global Radiation is used instead of UVB for H2SO4, it should be stated as it was done for OH. UVB is a fraction of Global Radiation, therefore a new coefficient should be used. Lu, Y., Yan, C., Fu, Y., Chen, Y., Liu, Y., Yang, G., Wang, Y., Bianchi, F., Chu, B., Zhou, Y., Yin, R., Baalbaki, R., Garmash, O., Deng, C., Wang, W., Liu, Y., Petäjä, T., Kerminen, V.-M., Jiang, J., Kulmala, M., and Wang, L.: A proxy for atmospheric daytime gaseous sulfuric acid concentration in urban Beijing, Atmos. Chem. Phys., 19, 1971–1983, https://doi.org/10.5194/acp-19-1971-2019, 2019.

**Response**: Thanks for the constructive comments, we re-calculated proxy $H_2SO_4$ by Lu et al., (2019) and did the reanalysis.

In the new calculation of $[H_2SO_4]$ in Beijing, we chose proxy equation number 2 (Eq. 1) and 7 (Eq. 2) as recommended by Lu et al. (2019), to represent the simplest and most accurate method, respectively.

$$[H_2SO_4] = 280.05 \times UVB^{0.14} \times [SO_2]^{0.40} \qquad (1)$$
$$[H_2SO_4] = 0.0013 \times UVB^{0.13} \times [SO_2]^{0.40} \times CS^{-0.17} \times ([O_3]^{0.44} + [NO_x]^{0.41}) \quad (2)$$

And the UVB was derived by 0.008% × Glob_R, based on the previous study that the monthly average of the ratio of UVB to global radiation (Glob_R) ranged from 0.007 to 0.017% in Beijing (Hu et al., 2013). The average ratio of January and February (0.008%) was applied.

However, the results derive by N2 and N7 method (Lu et al., 2019) showed a clear difference, indicating the large uncertainty of the proxy method. And for several NPF events, the elevated concentration of sulfuric acid was not observed by N2 and N7 method. The main reason could be the role of *CS* was underestimated. In the previous study in Beijing (Lu et al., 2019), the covariance of *CS* and $SO_2$ was found (*correlation coefficient R*=0.83) that offset the dependence of sulfuric acid on *CS*. However, during the measurement in our study, a special period of emission sharply decreased, *R* is 0.45 between $SO_2$ and *CS*. As a compromise, we also referred the proxy method develop in boreal forest (Petäjä et al., 2009). The average value of three proxy method was applied

to analyze the variation of sulfuric acid and its relationship with NPF events.

[Figure]

Fig. S5. The sulfuric acid concentrations derived by different proxy equations. The red and orange lines indicate the result by N2 and N7 method by Lu et al., 2019, and blue line indicates the method recommend by Petiäjä et al., 2009.

2. I didn't found the OH proxy used here in Petäjä et al (2009). Please make sure the right reference is cited.
**Response**: the correct citation should be "Nieminen, T., Keronen, P., Asmi, A., Petäjä, T., maso, M. D., Kulmala, M. and Kerminen, V.-M.: Trends in atmospheric new-particle formation: 16 years of observations in a boreal-forest environment, Boreal Envrion. Res., 19 (suppl. B): 191–214, 2014."

3. There are spelling mistakes and grammar errors, try to find an english expert to fix them all, eg: The number concentration of Aitken mode particles (~25-100nm) should also decreased as expected-> The number concentration of Aitken mode particles (~25-100nm) decreased as expected or The number concentration of Aitken mode particles (~25-100nm) decreased as they should.
**Response**: The spelling and grammar have checked through the manuscript, which has been language edited by the English native speakers. The sentence that reviewer mentioned has been revised to be "The number concentration of Aitken mode particles (~25-100 nm), which is related with the traffic emission (Deventer et al., 2018) is also expected to decrease."

4. Use the term oxidizing capacity consistently, replace all "oxidization capacity"
**Response**: It has been revised as the reviewer suggested.

---

## Author Response (AR2)

Response to reviewer's comments

The paper has been improved significantly, now I see that the story is mainly about nucleated particles which can grow into accumulation mode and contribute to PM2.5 mass concentration. Regarding this, there are still a few things I'm not sure I have understood well.

***Response***: We appreciate the reviewer for reviewing our manuscript and for the valuable suggestions, which we will address point by point in the following.

1. You mentioned in the abstract that CS declined during LCD period, while accumulation mode particle number and $PM_{2.5}$ concentration have both increased. This is not very usual, could you provide some information. Is it due to very low concentration of Aitken mode particles?

   ***Response***: The mean size distribution of *CS* was calculated and it showed the *CS* was mainly contributed by the accumulation mode particles (100-1000 nm). The accumulated particles can contribute to approximately 85% to the total *CS* in sub-micron size. Although the Aitken mode had larger number concentration, the contribution to *CS* and $PM_{2.5}$ mass concentration was minor as the small particle size.

[Figure]

Fig. 1. The size distribution of condensation sink (*CS*) during the measurement. The solid line and error bars represent the mean value and standard deviation.

2. You are referring to Guo et al.,2014, where they first came up to the idea that nucleated particles could contribute substantially to PM2.5 concentration in polluted area. However, is it still valid in this case? As already mentioned in question 1, Aitken mode particles have decreased. Why do nucleation mode increase and grew, but skipped the Aitken mode and entered accumulation directly? Maybe it would be worthy to focus on a single cases, since statistics on 2 months could result in misleading numbers.

   ***Response***: A specific case occurred on Feb 4th-14th was analyzed to discuss the influence of NPF event on the elevated $PM_{2.5}$ mass concentration. For this case, the Aitken mode particles on Feb 4th-6th increased due to the NPF event, and then decreased as the particles grow into the accumulation mode. As mentioned above, the accumulation mode was the major contributor to the $PM_{2.5}$ mass concentration. It was also reported by Kulmala et al (2021) that 65% of haze particles on hazy days were resulted from NPF based on one-year dataset in Beijing. However,

as the reviewer questioned, more robust confidence and quantitative evaluation of the NPF influence on the air pollution should be conducted by model work and long-term measurement in the future work. In the abstract, we addressed that the specific pollution case related with the NPF event could occur under the unfavorable meteorological conditions.

Ref:

Kulmala, M., Dada, L., Daellenbach, K. R., Yan, C., Stolzenburg, D., Kontkanen, J., Ezhova, E., Hakala, S., Tuovinen, S., Kokkonen, T. V., Kurppa, M., Cai, R., Zhou, Y., Yin, R., Baalbaki, R., Chan, T., Chu, B., Deng, C., Fu, Y., Ge, M., He, H., Heikkinen, L., Junninen, H., Liu, Y., Lu, Y., Nie, W., Rusanen, A., Vakkari, V., Wang, Y., Yang, G., Yao, L., Zheng, J., Kujansuu, J., Kangasluoma, J., Petaja, T., Paasonen, P., Jarvi, L., Worsnop, D., Ding, A., Liu, Y., Wang, L., Jiang, J., Bianchi, F. and Kerminen, V. M.: Is reducing new particle formation a plausible solution to mitigate particulate air pollution in Beijing and other Chinese megacities?, Faraday Discuss, 226: 334-347, DOI: 10.1039/d0fd00078g, 2021.

**Minor comments**:

The group of organics called organics was even still not discovered by the Atkinsen paper. I would suggest to cite the Ehn et al.(2012) and Bianchi et al.(2019) paper when you include HOMs, but what I would prefer is to delete all terms called HOMs and change them to oxidized VOCs or oxidized organics, since that is more exactly what is disscused here.

*Response*: The terms "HOMs" in the manuscript have been replaced by "oxidized VOCs" as reviewer recommend. The sentence has been revised to "The major pathways of oxidized VOCs formation are the oxidation by $O_3$, OH and $NO_3$ radicals (Ehn et al., 2012)".

Writing has also improved, but please continue to check typos like "adpotted" and correct them.

*Response*: The spelling and grammar have been checked and corrected all through the manuscript.